# MicroRNA156 conditions auxin sensitivity to enable growth plasticity in response to environmental changes in *Arabidopsis*

Qing Sang [1,2,3], Lusheng Fan[3], Tianxiang Liu [3], Yongjian Qiu [3,4], Juan Du [3], Beixin Mo [1,2], Meng Chen [3] ✉ & Xuemei Chen [3,5] ✉

MicroRNAs (miRNAs) play diverse roles in plant development, but whether and how miRNAs participate in thermomorphogenesis remain ambiguous. Here we show that HYPONASTIC LEAVES 1 (HYL1)−a key component of miRNA biogenesis−acts downstream of the thermal regulator PHYTOCHROME INTERACTING FACTOR 4 in the temperature-dependent plasticity of hypocotyl growth in *Arabidopsis*. A *hyl1-2* suppressor screen identified a dominant *dicer-like1* allele that rescues *hyl1-2*'s defects in miRNA biogenesis and thermoresponsive hypocotyl elongation. Genome-wide miRNA and transcriptome analysis revealed microRNA156 (miR156) and its target *SQUAMOSA PROMOTER-BINDING-PROTEIN-LIKE 9* (*SPL9*) to be critical regulators of thermomorphogenesis. Surprisingly, perturbation of the miR156/*SPL9* module disengages seedling responsiveness to warm temperatures by impeding auxin sensitivity. Moreover, miR156-dependent auxin sensitivity also operates in the shade avoidance response at lower temperatures. Thus, these results unveil the miR156/*SPL9* module as a previously uncharacterized genetic circuit that enables plant growth plasticity in response to environmental temperature and light changes.

Phenotypic plasticity in response to temperature changes is critical for the survival of plants in diverse geographical environments and a constantly evolving global climate[1]. A moderate increase in the ambient temperature of only a few degrees can elicit dramatic adaptive responses in plant development, growth, metabolism, and immunity, which are collectively called thermomorphogenesis[2,3]. The need for a molecular understanding of the mechanism underpinning thermomorphogenesis has become imminent to predict and mitigate the impacts of an altered climate on species distribution, community composition, and crop productivity[4,5].

The embryonic stem (hypocotyl) of *Arabidopsis thaliana* (*Arabidopsis*) is a well-established model used to interrogate the mechanism

of growth plasticity in response to environmental cues. Warmer temperatures greatly accelerate hypocotyl growth[6]. Plants sense changes in ambient temperature via the red (R) and far-red (FR) photoreceptor phytochrome B (PHYB)[7,8]. PHYB monitors light quality, quantity, and periodicity through reversible photoconversions between two relatively stable conformers, an R-light-absorbing inactive Pr and an FR-light-absorbing active Pfr[9,10]. The active Pfr can also spontaneously revert to the inactive Pr in a light-independent process called thermal reversion. The rate of thermal reversion of PHYB becomes faster with temperature increases between 10 °C and 30 °C−i.e., warm temperatures destabilize the active Pfr−thereby making PHYB a thermosensor in addition to a photoreceptor[7,8]. As active PHYB inhibits hypocotyl

[1]Guangdong Provincial Key Laboratory for Plant Epigenetics, College of Life Sciences and Oceanography, Shenzhen University, Shenzhen 518060, China. [2]Key Laboratory of Optoelectronic Devices and Systems of the Ministry of Education and Guangdong Province, College of Optoelectronic Engineering, Shenzhen University, Shenzhen 518060, China. [3]Department of Botany and Plant Sciences, Institute for Integrative Genome Biology, University of California, Riverside, CA 92521, USA. [4]Present address: Department of Biology, University of Mississippi, Oxford, MS 38677, USA. [5]Present address: School of Life Sciences, Peking-Tsinghua Joint Center for Life Sciences, Peking University, Beijing 100871, China. ✉e-mail: meng.chen@ucr.edu; xuemei.chen@pku.edu.cn

elongation, the Pfr/Pr equilibrium of PHYB works as an adjustable thermal and photo switch to modulate hypocotyl elongation in response to temperature and light cues. Thus, warm temperatures act similarly to the FR light reflected in the shade of neighboring plants to attenuate the inhibition of hypocotyl growth by PHYB.

PHYB controls hypocotyl elongation by regulating the biosynthesis and signaling of phytohormones, such as auxin. PHYB interacts directly with a family of basic helix-loop-helix transcription factors, known as PHYTOCHROME-INTERACTING FACTORs (PIFs), to control their stability and transcriptional activity[11,12]. The growth plasticity of the hypocotyl is regulated mainly by PIF4, 5, and 7[13–16]. Warm temperatures promote the expression, accumulation, and activity of PIF4 and PIF7, which in turn activate genes associated with auxin biosynthesis and signaling[13–15,17,18]. The current model posits that warm temperatures, as well as shade, are perceived by PHYB primarily in the cotyledons and young leaves to enhance auxin production by inducing the expression of *YUCCA* (*YUC*) genes that encode flavin monooxygenase-like enzymes for auxin biosynthesis[19,20]. Subsequently, auxin is transported into the hypocotyl and acts collaboratively with other phytohormones, including brassinosteroids, to generate the cell elongation response[19–22].

MicroRNAs (miRNAs) are 20-to-24 nucleotide-long endogenous noncoding RNAs that regulate almost all aspects of plant development and growth[23]. *MIRNA* genes are first transcribed into long primary miRNA (pri-miRNA) transcripts, which are then processed into miRNA duplexes by the RNAase III family enzyme DICER-LIKE 1 (DCL1) and two accessory proteins, the dsRNA-binding protein HYPONASTIC LEAVES 1 (HYL1) and the C2H2 zinc finger protein SERRATE (SE)[24–26]. A miRNA duplex is then loaded primarily into ARGONAUTE 1 (AGO1), and one strand of the duplex is retained as the miRNA guide that targets AGO1 to the cognate target messenger RNA based on sequence complementarity, resulting in the degradation or translational repression of the target transcripts[27–31].

The role of miRNAs in thermomorphogenesis remains ambiguous. The interaction between miRNAs and thermomorphogenesis has been best explored in the context of floral induction. Shoot development can be divided into three phases: a juvenile vegetative phase, an adult vegetative phase, and a reproductive phase. miR156, which is expressed at very high levels in the juvenile phase and declines as the shoot develops, serves as an endogenous developmental regulator of both the juvenile-to-adult vegetative and the vegetative-to-reproductive phase transitions[32]. miR156 exerts its functions by repressing the expression of 10 *SQUAMOSA PROMOTER BINDING PROTEIN-LIKE* (*SPL*) transcription factors[33]. Elevated growth temperature accelerates flowering[34]. One proposed mechanism for this process is via the temperature-dependent regulation of miR156 accumulation. Several attempts were made to identify ambient temperature-responsive miRNAs in *Arabidopsis*. By comparing 10-day-old *Arabidopsis* seedlings grown at 16 °C and 23 °C, Lee et al. identified six temperature-responsive miRNAs, including four miRNAs (miR163, miR172, miR398, and miR399) upregulated and two (miR156 and miR169) downregulated by the warmer temperature[35,36]. The decrease in miR156 accumulation was attributed to reduced enzymatic efficiency in the processing of pri-miR156 into mature miR156 at the warmer temperature[37]. Supporting this model, the precision of the DCL1 activity appeared to be less robust and more reliant on the accessory proteins HYL1 and SE at higher temperatures[38]. Moreover, the thermal regulators PIF4 and PIF5 have been reported to directly suppress the expression of *MIR156* in shade conditions[39], implying that warm temperatures could also downregulate miR156 abundance at the transcriptional level. However, arguing against this model, the temperature-dependent accumulation of miR156 has not been observed consistently. In two other studies, the miR156 level was found to be either unchanged or even increased at warmer temperatures[38,40], suggesting the level of miR156 could be largely influenced by the growth conditions and developmental stages of the plants.

Circumstantial evidence suggests miRNAs may contribute to temperature-dependent plant growth plasticity. First, the thermoregulator PIF4 has been shown to interact with DCL1 and HYL1 to influence the biogenesis of miRNAs involved in hypocotyl growth regulation, raising the possibility that temperature signaling and miRNA biogenesis are connected directly[41]. Second, miRNAs could participate in thermomorphogenesis via the regulation of auxin signaling. Auxin is perceived by the TRANSPORT INHIBITOR RESPONSE 1/AUXIN RESPONSE F-BOX (TIR1/AFB) auxin co-receptors, which are the substrate recognition subunits of the CULLIN1-based E3 ubiquitin ligase SCF[TIR1/AFB]. Auxin binding to TIR1/AFB promotes the degradation of transcription repressors called AUXIN/INDOLE ACETIC ACID (Aux/IAA) proteins. The proteolysis of Aux/IAA proteins relieves their inhibition of AUXIN RESPONSE FACTORs (ARFs), leading to the transcription activation of downstream auxin-responsive genes[42,43]. The auxin-responsive genes include critical components of auxin signaling, such as the *Aux/IAA* genes encoding transcriptional repressors of the auxin response, as well as a large family of *SMALL AUXIN UP RNA* (*SAUR*) genes, whose products regulate the activity of plasma membrane H+-ATPases to promote apoplastic acidification and plasma membrane hyperpolarization in cell elongation[44]. Auxin signaling is the target of several miRNAs[45]. For instance, miR393 targets *TIR1/AFB*; miR160 and miR167 target *ARF10/16/17* and *AFR6/8*, respectively; and miR390 triggers the production of *TAS3*-derived trans-activating short-interfering RNAs (tasiRNAs) that negatively regulate *ARF2*, *ARF3*, and *ARF4*. However, whether these auxin-relevant miRNAs are involved in thermomorphogenesis remains unclear.

To determine whether and how miRNAs participate in thermomorphogenesis, here we utilized the *Arabidopsis* hypocotyl as a model to test whether mutants in miRNA biogenesis are impaired in the thermoresponsive hypocotyl growth. Using a combination of genetics, genomics, and molecular biology approaches, we demonstrate that miRNA biogenesis plays an essential role in thermomorphogenesis. Unexpectedly, we identified miR156 and its target *SPL9* as novel regulators of thermo-inducible hypocotyl elongation. Intriguingly, the miR156/*SPL9* module enables thermomorphogenesis by licensing auxin sensitivity. Moreover, miR156-dependent auxin sensitivity also operates in the shade avoidance response at lower temperatures. Together, our results unveil the miR156/*SPL9* module as a previously uncharacterized developmental pathway that enables plant's growth plasticity in response to environmental temperature and light changes.

## Results

### HYL1 is required for thermomorphogenesis

To explore the function of miRNAs in thermomorphogenesis, we first examined the PHYB-mediated thermoresponsive hypocotyl elongation between 21 °C and 27 °C in miRNA biogenesis mutants *dcl1-20*, *hyl1-2*, and *se-1*. To avoid the complex influence of the blue light photoreceptor CRY1 on thermomorphogenesis[14,46], we measured the daytime temperature response under continuous monochromatic R light[14,17]. Because thermomorphogenesis is mediated prominently by three bHLH transcription factors, PIF4, PIF5, and PIF7[13–16], we used *pif4-2* and *pif457* as negative controls. Among the three miRNA biogenesis mutants, only *hyl1-2* showed a dramatic defect in the temperature response, comparable to that of *pif4-2* (Fig. 1a, b), while *dcl1-20* exhibited a normal response like that of Col-0, and *se-1* had a moderately reduced response (Fig. 1a, b). The discrepancies among the miRNA biogenesis mutants can be explained by the fact that only *hyl1-2* is a null allele[26,47], whereas neither *dcl1-20*[48] nor *se-1*[49] is a null mutant because their null mutants are embryonic lethal[50,51]. However, it was equally possible that the phenotype of *hyl1-2* was attributable to a unique function of HYL1 independent of miRNA biogenesis.

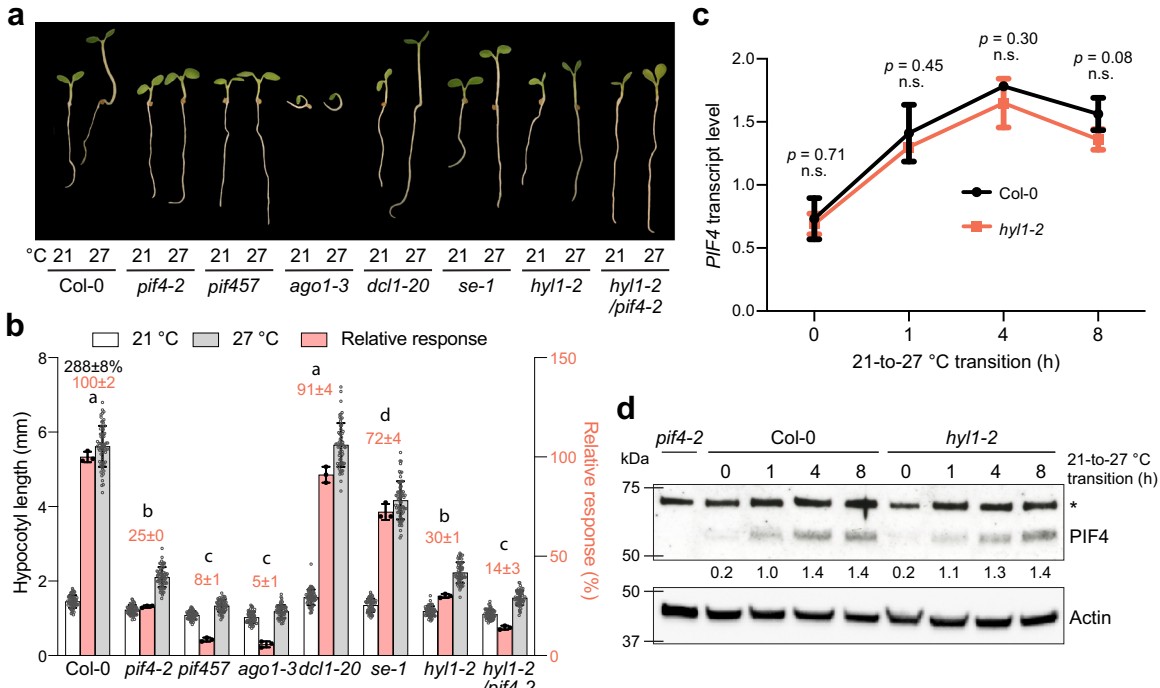

**Fig. 1 | HYL1 is required for thermomorphogenesis. a** Representative images of 4-d-old Col-0, *pif4-2*, *pif457*, *ago1-3*, *dcl1-20*, *se-1*, *hyl1-2*, and *hyl1-2/pif4-2* seedlings grown under 50 μmol m$^{-2}$ s$^{-1}$ R light at either 21 °C or 27 °C. **b** Hypocotyl length measurements of the seedlings in (**a**) and their relative responses to the higher temperature. The open and gray bars represent hypocotyl length measurements at 21 °C and 27 °C, respectively. Error bars for the hypocotyl measurements represent s.d. (*n* = 60 seedlings). The black number above the Col-0 columns represents the percent increase in hypocotyl length (mean ± s.d., *n* = 3 biological replicates) at 27 °C compared with 21 °C. The pink bars show the relative response, which is defined as the hypocotyl response to 27 °C of a mutant relative to that of Col-0 (set at 100%). Pink numbers show the mean ± s.d. values of the relative responses. Different lowercase letters above the bars denote statistically significant differences in the relative responses (ANOVA, Tukey's HSD, *p* < 0.01, *n* = 3 biological replicates). Error bars for the relative responses represent the s.d. of three biological replicates. The centers of all error bars indicate the mean values.

**c** qRT-PCR analysis of the *PIF4* transcript levels in 4-d-old Col-0 and *hyl1-2* seedlings grown under 50 μmol m$^{-2}$ s$^{-1}$ R light at the indicated time points during the 21-to-27 °C transition. The transcript levels were quantified relative to those of *PP2A*. Error bars represent the s.d. of three biological replicates. The centers of the error bars represent the mean values. Statistical significance was analyzed using a two-tailed Student's t-test. **d** Immunoblot analyses of PIF4 protein levels in response to elevated temperature. Four-day-old Col-0 and *hyl1-2* seedlings grown at 50 μmol m$^{-2}$ s$^{-1}$ R light at 21 °C were transferred to 27 °C under the same light conditions for up to 8 h, and samples were collected at the indicated time points. The dark-grown *pif4-2* sample was used as a negative control. Actin was used as a loading control. The relative levels of PIF4, normalized to actin, are shown underneath the PIF4 immunoblots. The asterisk indicates nonspecific bands. The experiment was repeated three times with similar results. The source data underlying the hypocotyl measurements in (**b**), the qRT-PCR analysis in (**c**), and the immunoblots in (**d**) are provided in the Source Data file.

To test the genetic relationship between HYL1 and PIF4 in thermomorphogenesis, we generated a *hyl1-2/pif4-2* double mutant. The *hyl1-2/pif4-2* mutant was slightly less responsive to the warm temperature than the single *hyl1-2* and *pif4-2* mutants (Fig. 1a, b), suggesting that HYL1 and PIF4 work in both overlapping and parallel pathways. One possibility is that the function of HYL1 is required for the action of multiple PIFs, as the phenotype of *hyl1-2/pif4-2* was similar to *pif457* (Fig. 1a, b). However, we could not exclude that HYL1 is also involved in PIF-independent pathways. The *hyl1-2* mutant exhibited a normal response in the warm-temperature-induced accumulation of *PIF4* mRNA and protein (Fig. 1c, d), suggesting that HYL1 is dispensable for temperature sensing and the early signaling steps leading to PIF4 accumulation.

**miRNA biogenesis plays an essential role in thermomorphogenesis**

To test whether *hyl1-2*'s defect in thermomorphogenesis was due to its deficiency in miRNA biogenesis, we examined the temperature response in a null allele of *ago1*, *ago1-3*, which is expected to abolish global miRNA functions[52]. Intriguingly, *ago1-3* almost completely lost the warm temperature response, displaying a phenotype similar to *pif457* (Fig. 1a, b). These results strongly support an important role of miRNAs in thermomorphogenesis. However, because *ago1-3* exhibits severe and pleiotropic developmental defects, the lack of the

thermoresponsive hypocotyl response in *ago1-3* could be due to a general developmental deficiency as opposed to a specific block in temperature signaling.

To further determine the role of miRNAs in thermomorphogenesis, we performed a forward genetic screen for suppressor mutations that could restore the hypocotyl elongation response at 27 °C in *hyl1-2*. We mutagenized *hyl1-2* seeds with ethyl methanesulfonate (EMS) and screened for suppressor mutants that were taller than *hyl1-2* at 27 °C. To eliminate mutants with a long hypocotyl at both high and low temperatures, such as a *phyB* mutant, we conducted a secondary screen to search for mutants that had a similar hypocotyl length as *hyl1-2* at the low temperature—i.e., we screened for mutants that were taller than *hyl1-2* at 27 °C but not 21 °C. Three *hyl1-2* suppressor (*hs*) mutations, named *hs400*, *hs470*, and *hs471*, completely restored the thermoresponsive hypocotyl response in *hyl1-2* (Fig. 2a, b). We backcrossed *hs400/hyl1-2* to *hyl1-2* to generate a B2 (backcross 2 generation) mapping population. The suppressor phenotype segregated in about 75% of the B2 seedlings, indicating that the suppressor mutation was dominant. Then, we utilized Illumina sequencing to sequence the DNA pooled from at least 96 *hyl1-2*-like B2 plants—i.e., seedlings with a short hypocotyl at 27 °C—as well as the DNA from homozygous *hs400/hyl1-2* in the M3 generation. Using a point mutation mapping pipeline called SIMPLE[53], we identified in *hs400/hyl1-2* a mutation in the *DCL1* locus (AT1G01040), which leads to a proline-to-serine substitution at

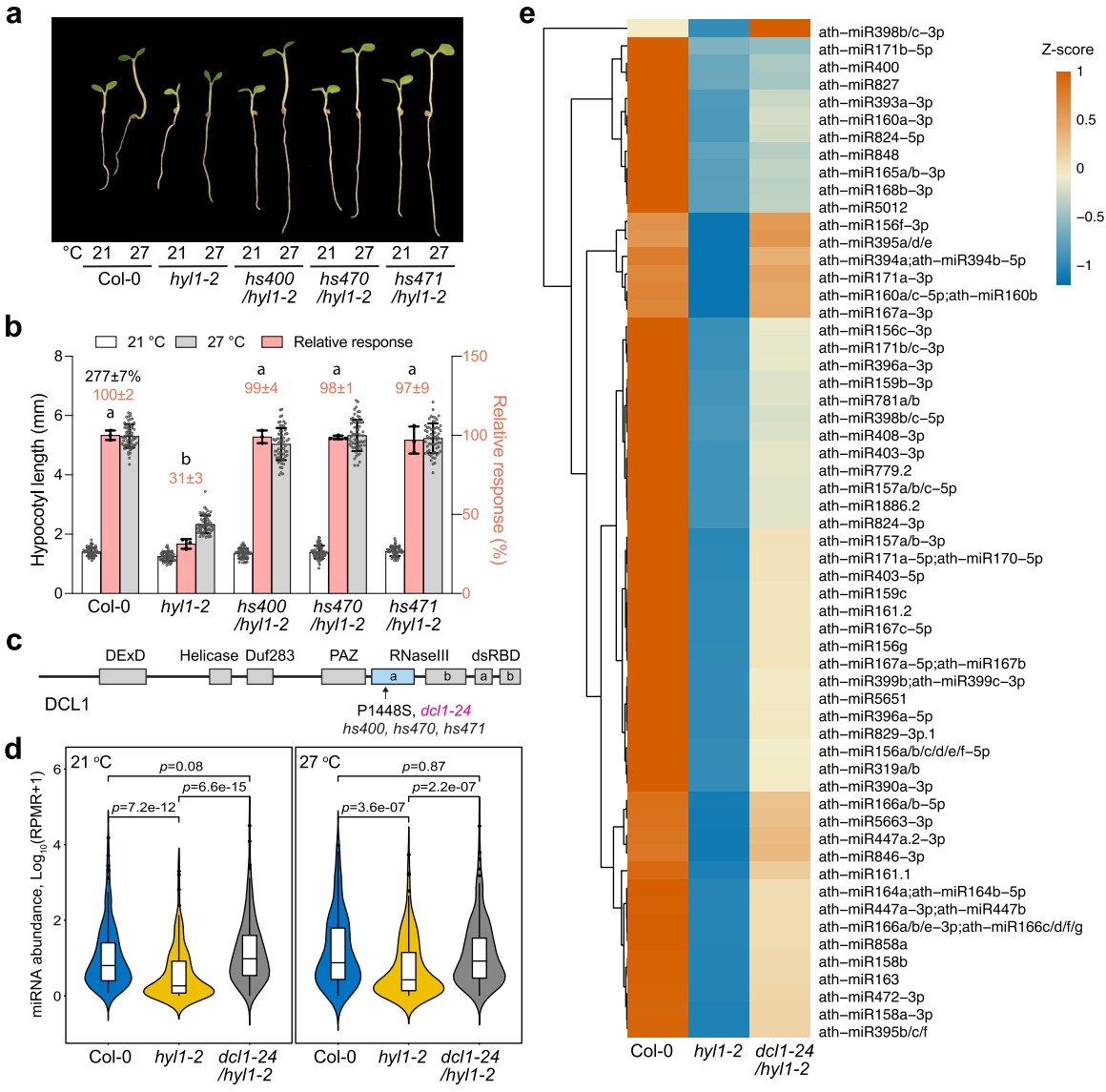

**Fig. 2 | MicroRNA biogenesis is essential for thermomorphogenesis.**
**a** Representative images of 4-d-old Col-0, *hyl1-2*, *hs400/hyl1-2*, *hs470/hyl1-2*, and *hs471/hyl1-2* seedlings grown under 50 μmol m⁻² s⁻¹ R light at either 21 °C or 27 °C.
**b** Hypocotyl length measurements of the seedlings in (**a**) and their relative responses to the warm temperature. The open and gray bars represent hypocotyl length measurements at 21 °C and 27 °C, respectively. Error bars for the hypocotyl measurements represent s.d. (*n* = 60 seedlings). The black number above the Col-0 columns represents the percent increase in hypocotyl length (mean ± s.d., *n* = 3 biological replicates) at 27 °C compared with 21 °C. The pink bars show the warm-temperature response of a mutant relative to that of Col-0 (set at 100%). Pink numbers show the mean ± s.d. values of the relative responses. Different lowercase letters above the bars denote statistically significant differences in the relative responses (ANOVA, Tukey's HSD, *p* < 0.01, *n* = 3 biological replicates). Error bars for

the relative responses represent the s.d. of three biological replicates. The centers of all error bars indicate the mean values. **c** Schematic illustration of the domain structure of DCL1 and the position of the suppressor *dcl1-24* mutation. **d** Violin plots showing the relative global miRNA levels in Col-0, *hyl1-2*, and *dcl1-24/hyl1-2* at 21 °C (left) and 27 °C (right). The miRNA levels were quantified relative to the respective reads from small RNA fragments derived from 45S rRNA. RPMR, reads per million of rRNA reads. In the box and whisker plots, the boxes represent the 25% to 75% quantiles, and the bars are equal to the median. Statistical significance was analyzed using a two-tailed Student's t-test, *n* = 3 independent biological replicates. **e** Heatmap showing the relative levels of the 56 miRNAs downregulated in *hyl1-2* and rescued in *dcl1-24/hyl1-2* at 27 °C. The source data underlying the hypocotyl measurements in (**b**) are provided in the Source Data file.

amino acid 1448 (P1448S) in DCL1 (Fig. 2c). Interestingly, this exact mutant allele was previously isolated as *dcl1-24* from a different *hyl1-2* suppressor screen based on the rescue of the leaf hyponasty phenotype in adult *hyl1-2* plants[54]. We then sequenced the *DCL1* locus of *hs470/hyl1-2* and *hs471/hyl1-2* and found that these two suppressor mutants also carried the same mutation as in the previously described *dcl1-24*. The *hs400/hyl1-2* mutant, designated *dcl1-24/hyl1-2* hereafter, was used for the subsequent analysis.

The P1448S mutation lies in the RNaseIIIa domain of DCL1 (Fig. 2c). The RNaseIIIa domain works with the helicase domain to exert an autoinhibitory function on DCL1's catalytic activity, which can be

activated by HYL1[54]. It was suggested that the P1448S substitution in DCL1, and also five other mutations in the RNaseIIIa and helicase domains, could suppress the autoinhibitory function of these two domains, thereby releasing DCL1's catalytic activity in the absence of HYL1[54,55]. To confirm that *dcl1-24* could rescue the miRNA biogenesis defects in *hyl1-2* in our experimental settings, we performed small RNA sequencing to quantify the genome-wide miRNA levels in Col-0, *hyl1-2*, and *dcl1-24/hyl1-2* seedlings grown at both 21 °C and 27 °C. As expected, the levels of miRNAs decreased globally in *hyl1-2* compared to Col-0, and this phenotype of *hyl1-2* was largely recovered in *dcl1-24/hyl1-2* at both temperatures (Fig. 2d and Supplementary Data 1). At 27 °C, 65

miRNAs were downregulated significantly ($p < 0.01$) by twofold in *hyl1-2* compared to Col-0; 56 of them were rescued to levels similar (within twofold) to those in Col-0 in *dcl1-24/hyl1-2* (Fig. 2e and Supplementary Data 1). Thus, the *dcl1-24/hyl1-2* suppressor mutant provides genetic evidence supporting that the thermomorphogenesis defect in *hyl1-2* is due to the deficiency in miRNA biogenesis.

### Thermomorphogenesis requires miR156

As one possible mechanism linking miRNAs and thermomorphogenesis is through temperature-responsive miRNAs[35,37,40], we set out to identify miRNAs upregulated by warm temperatures in Col-0 and *dcl1-24/hyl1-2*. The levels of five miRNAs increased statistically significantly by twofold at 27 °C compared with 21 °C in Col-0; however, none of them increased in abundance at 27 °C in *dcl1-24/hyl1-2* (Fig. 3a and Supplementary Data 1). These results suggest that the abundance of the miRNA(s) promoting thermomorphogenesis might not be significantly altered by temperature in our experimental settings.

We then asked whether we could identify the causal miRNA(s) required for thermomorphogenesis by analyzing the expression of miRNA targets. To that end, we performed RNA-seq analysis to examine the expression of genome-wide miRNA targets in Col-0, *hyl1-2*, and *dcl1-24/hyl1-2* at 27 °C. Interestingly, only 14 of the 140 annotated miRNA targets[56] in *Arabidopsis* were upregulated in *hyl1-2* and rescued in *dcl1-24/hyl1-2*; these genes corresponded to 8 cognate miRNAs (Fig. 3b, c and Supplementary Data 2), 6 of which were among the 56 miRNAs downregulated in *hyl1-2* and rescued in *dcl1-24/hyl1-2* (Figs. 3c and 2e). One plausible candidate was miR156, because four of the ten miR156-regulated *SPL*s—*SPL3*, *SPL5*, *SPL6*, and *SPL9*—were upregulated in *hyl1-2* and rescued in *dcl1-24/hyl1-2* (Fig. 3c). We then examined the thermal response in a transgenic line constitutively expressing a miR156 target mimicry (*MIM156*) to block miR156 activity[57]. Intriguingly, the *MIM156* seedlings exhibited a greatly reduced hypocotyl response to warmer temperatures (Fig. 3d, e).

Then, we asked whether miR156 mediates thermomorphogenesis by repressing the *SPL*s. The ten miR156-regulated *SPL*s can be classified into three functionally distinct groups: (1) *SPL2*, *SPL9*, *SPL10*, *SPL11*, *SPL13*, and *SPL15* control both the juvenile-to-adult vegetative transition and the vegetative-to-reproductive transition; (2) *SPL3*, *SPL4*, and *SPL5* do not contribute to the vegetative and vegetative-to-reproductive developmental transitions, instead mainly promoting floral meristem identity transition; (3) *SPL6* does not play a major role in shoot morphogenesis[33]. To gauge the roles of the *SPL*s in miR156-mediated thermomorphogenesis, we obtained transgenic lines expressing either miR156-sensitive (*sSPL*) or miR156-resistant (*rSPL*)[33] version of the *SPL*s and examined the difference in their hypocotyl responses to warm temperatures. Among the ten *SPL*s, *SPL9* was the only gene whose miR156-resistant line (*rSPL9*) displayed a more than twofold decrease in the warm temperature response compared to the miR156-sensitive line (*sSPL9*), indicating that SPL9 is the primary SPL inhibiting thermoresponsive hypocotyl elongation under our experimental conditions (Fig. 3d, e and Supplementary Fig. 1a). The thermoresponse defect of *rSPL9* was similar to that of *MIM156* (Fig. 3d, e). Both *rSPL2* and *rSPL3* also showed a slight decrease in the hypocotyl thermal response compared to their respective *sSPL* lines, however, their phenotypes were much weaker compared to that of *rSPL9* (Supplementary Fig. 1a). *rSPL10*, *rSPL11*, and *rSPL15* showed an enhanced thermal response compared to their corresponding *sSPL* lines, therefore, these *SPL*s may promote hypocotyl elongation at warm temperatures (Supplementary Fig. 1a). The *spl9-4*, *spl2/9/10/11/13/15*, and *spl3/4/5* mutants responded to warm temperatures similarly to the wild-type (Supplementary Fig. 1b), suggesting that the function of *SPL9* (and also other *SPL*s) was effectively repressed by miR156 in our assay conditions. Together, these results demonstrate that miR156 promotes thermoresponsive hypocotyl growth by repressing mainly *SPL9*.

### miR156 allows the proper regulation of auxin-responsive genes by warm temperatures

Thermomorphogenesis is mediated by the PIF-dependent activation of genes associated with auxin biosynthesis and signaling. To determine the role of miR156 in thermomorphogenesis, we examined genome-wide temperature-responsive genes in Col-0, *pif457*, *MIM156*, *hyl1-2*, and *dcl1-24/hyl1-2* by RNA-seq analysis (Supplementary Data 2). Gene ontology (GO) analysis of transcriptomic changes in Col-0 between 21 °C and 27 °C identified genes enriched in six warm-temperature-induced biological processes, including cellular response to decreased oxygen levels, cellular response to oxygen levels, cellular response to hypoxia, response to heat, response to water, and response to auxin (Fig. 4a and Supplementary Data 2). We compared the warm-temperature-induced biological processes among Col-0, *pif457*, *MIM156*, *hyl1-2*, and *dcl1-24/hyl1-2* using clusterProfiler[58]. Interestingly, only the enrichment of genes in response to auxin disappeared in *pif457*, confirming that PIFs function primarily to invoke the auxin response (Fig. 4a). *MIM156* showed a similar gene enrichment pattern as *pif457*: the GO category of auxin-responsive genes was no longer enriched, although the number of genes associated with response to oxygen levels was also lower in *MIM156* than in Col-0 and *pif457* (Fig. 4a). These results suggest that similar to PIFs, miR156 was also required for the proper induction of auxin-responsive genes by warm temperatures. In contrast, *hyl1-2* had complex effects on all warm-temperature-induced categories, including, surprisingly, an enhanced enrichment of auxin-responsive genes (Fig. 4a). The GO enrichment pattern of *dcl1-24/hy1-2* was reversed back to being Col-0-like, consistent with the rescue of the temperature responsive phenotype. We presumed that the drastic changes in the GO enrichment pattern in *hyl1-2* were due to the global reduction of miRNAs, whereas the changes in *MIM156* reflected the specific role of miR156 in the regulation of auxin-responsive genes.

We then examined the expression of the 317 genes in the GO category of auxin-responsive genes (GO:00009733). Among these auxin-responsive genes, 31 were upregulated in Col-0 at 27 °C compared with 21 °C (Fig. 4b). The overall temperature responsiveness of the 31 auxin-responsive genes appeared to be reduced similarly between *pif457* and *MIM156* (Fig. 4c). Specifically, the expression of 18 temperature-induced auxin-responsive genes depended on both PIFs and miR156 (Fig. 4b). These included the established warm-temperature-induced auxin signaling genes, such as members of the *SAUR19* and *SAUR26* subfamilies, including *SAUR19*, *SAUR20*, *SAUR22-24*, *SAUR27*, and *SAUR28* as well as *Aux/IAA* genes, including *IAA19*, *IAA29*, *IAA5*, and *IAA34* (Fig. 4c)[59–61]. Consistent with the GO enrichment analysis, only 7 of the 31 warm-temperature-induced auxin-responsive genes were also dependent on HYL1—i.e., a large fraction of them remained inducible by warm temperatures in *hyl1-2* (Fig. 4b, c). However, a closer comparison of the auxin-responsive genes between *dcl1-24/hyl1* and *hyl1-2* revealed that many of the highly expressed auxin responsive-genes, e.g., *SAUR20*, *SAUR21*, *SAUR23*, and *PIR3*, were downregulated in *dcl1-24/hyl1*. These results suggest that the auxin-responsive genes might not be properly coordinated, and some of them could be overexpressed in *hyl1-2*. Consistent with the idea of the uncoordinated expression of auxin-responsive genes in *hyl1-2*, 20 additional auxin-responsive genes, which were not induced by warm temperatures in Col-0, became upregulated in *hyl1-2*, whereas only 6 such genes were upregulated in *MIM156* (Fig. 4b), suggesting that a deficiency in multiple miRNAs could disrupt the proper regulation of auxin-responsive genes in *hyl1-2*. We also examined the expression of the *YUC* genes that participate in warm-temperature-induced auxin biosynthesis[19,20]. Two of these genes, *YUC8* and *YUC9*, were expressed at detectable levels, and, as expected, both were upregulated by warm temperature in Col-0 but not in *pif457* (Fig. 4d). *YUC8* and *YUC9* remained to be upregulated or highly expressed in *MIM156* and *hyl1-2* at 27 °C (Fig. 4d), implying that miR156 plays a smaller role in

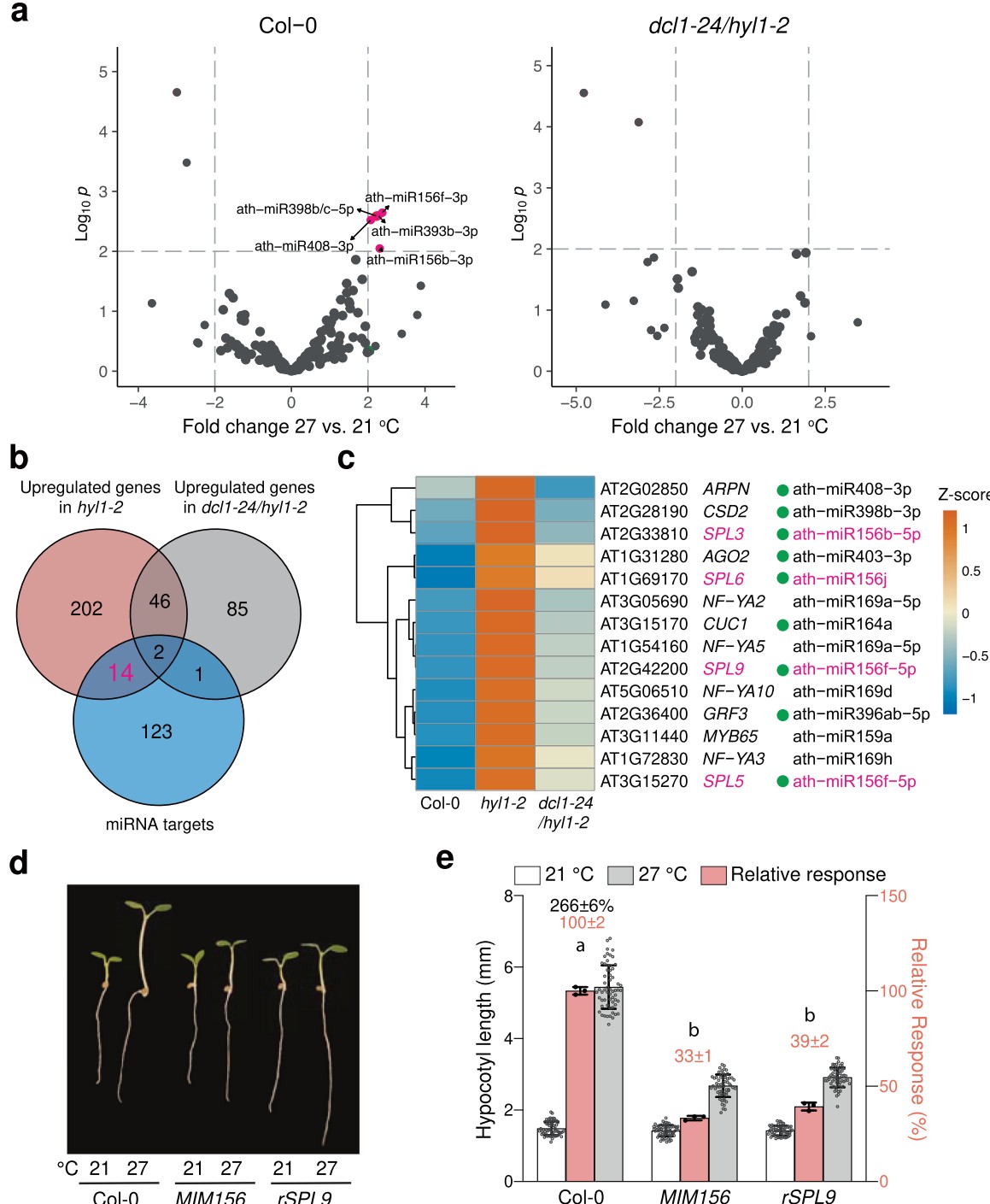

**Fig. 3 | Thermomorphogenesis requires miR156. a** Volcano plots showing differentially accumulated (two-tailed $p < 0.01$, and fold change >2) miRNAs in 4-d-old Col-0 and *dcl1-24/hyl1-2* seedlings grown under 50 μmol m$^{-2}$ s$^{-1}$ R light between 21 °C and 27 °C. Statistical significance was analyzed using multiple t-tests with correction for multiple comparisons. **b** Venn diagram depicting that 14 miRNA targets were upregulated in *hyl1-2* and rescued in *dcl1-1/hyl1-2*. **c** Heatmap showing the relative expression levels of the 14 *HYL1*-regulated miRNA targets identified in (**b**) in 4-d-old Col-0, *hyl1-2*, and *dcl1-24/hyl1-2* seedlings grown under 50 μmol m$^{-2}$ s$^{-1}$ R light at 27 °C. The green dots indicate that the corresponding miRNAs were among the 56 downregulated miRNAs in *hyl1-2* and rescued in *dcl1-24/hyl1-2*, as shown in Fig. 2e. miRNA156 and *SPLs* are highlighted in magenta. **d** Representative images of 4-d-old Col-0, *MIM156*, and *rSPL9* seedlings grown under 50 μmol m$^{-2}$ s$^{-1}$ R

light at either 21 °C or 27 °C. **e** Hypocotyl length measurements of the seedlings in (**d**) and their relative responses to the higher temperature. Error bars represent s.d. ($n = 60$ seedlings). The black number above the Col-0 columns represents the percent increase in hypocotyl length (mean ± s.d., $n = 3$ independent biological replicates) at 27 °C compared with 21 °C. The pink bars show the warm-temperature response of a mutant relative to that of Col-0 (set at 100%). Pink numbers show the mean ± s.d. of the relative responses. Different lowercase letters above the bars denote statistically significant differences in the relative responses (ANOVA, Tukey's HSD, $p < 0.01$, $n = 3$ biological replicates). Error bars for the relative responses represent the s.d. of three biological replicates. The centers of the error bars indicate the mean. The source data underlying the hypocotyl measurements in (**e**) are provided in the Source Data file.

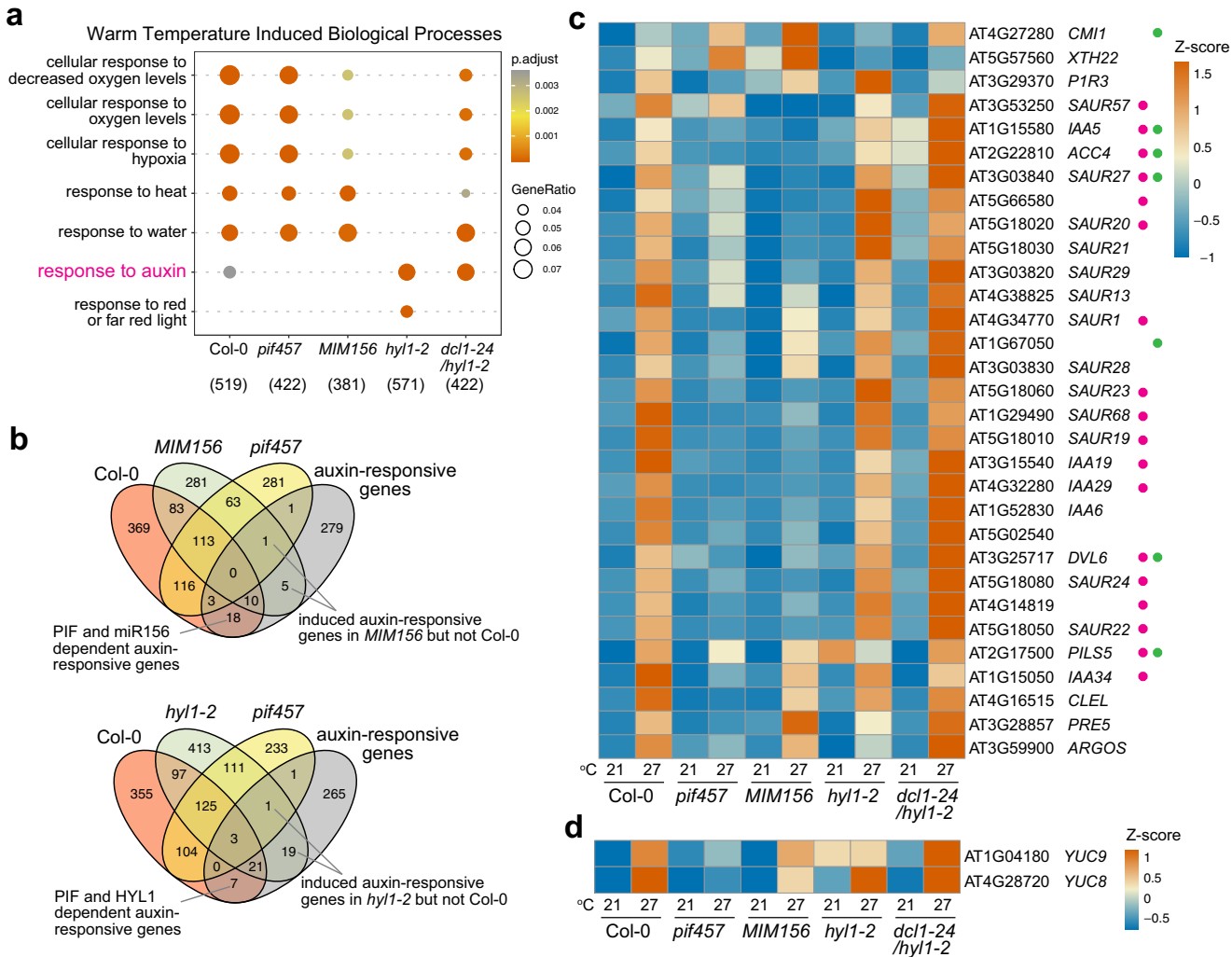

**Fig. 4 | MiR156 potentiates the proper regulation of auxin-responsive genes by warm temperatures. a** GO enrichment analysis of the top warm-temperature-induced biological processes in Col-0, *pif457*, *MIM156*, *hyl1-2*, and *dcl1-24/hyl1-2*. A comparison of the enriched biological processes among the genetic backgrounds was performed using clusterProfiler[58], with the strict cutoff of *p* < 0.01 and FDR < 0.05. The numbers in parentheses indicate the number of warm-temperature-induced genes in each genotype. **b** Venn diagrams showing the numbers of warm-temperature-induced auxin-responsive genes (GO:0009733) in Col-0, *pif457*, *hyl1-2*, and *MIM156*. **c** Heatmap showing the relative expression levels of the 31 warm-temperature-upregulated auxin-responsive genes in Col-0, *pif457*, *MIM156*, *hyl1-2*, and *dcl1-24/hyl1-2* at 21 °C and 27 °C. The magenta dots indicate the 18 auxin-responsive genes that are dependent on both PIFs and miR156. The green dots indicate the 7 auxin-responsive genes that are dependent on both PIFs and HYL1. **d** Heatmap showing the relative expression levels of *YUC8* and *YUC9* in Col-0, *pif457*, *MIM156*, *hyl1-2*, and *dcl1-24/hyl1-2* at 21 °C and 27 °C.

regulating auxin biosynthesis. Taken together, these RNA-seq results support the conclusion that miR156 is required for the proper regulation of auxin-responsive gene expression by warm temperatures.

## Perturbation of miR156/SPL9 impedes auxin responsiveness at warm temperatures

The current model of thermomorphogenesis posits that the warm-temperature-induced accumulation of PIF4 and PIF7 promotes auxin biosynthesis in cotyledons and the transport of auxin from the cotyledons to the hypocotyl, where auxin works together with other hormones, particularly brassinosteroids (BRs) to trigger cell elongation in the hypocotyl[19–22]. Auxin responsiveness also depends on BR signaling, as blocking BR synthesis or signaling alters auxin-responsive gene expression and diminishes the hypocotyl's responsiveness to auxin[62]. Therefore, miR156 could modulate auxin-responsive gene expression either directly or indirectly via BR signaling. Mutants in upstream components, such as PIF4 and auxin signaling, can be distinguished from downstream mutants in BR signaling by their distinct responses to exogenous BR treatments or auxin treatments. For example,

treating *pif4-2* and auxin-deficient mutants with exogenous brassinolide rescues their hypocotyl growth retardation at warm temperatures; in contrast, exogenous auxin cannot rescue the growth defects of BR biosynthesis and signaling mutants[21]. We used the same approaches to determine if miR156 is required for proper hypocotyl responsiveness to exogenous BR and auxin treatments. The exogenous application of 100 nM brassinolide at 27 °C had little effect on Col-0 but rescued the hypocotyl growth retardation of the BR biosynthesis mutant *det2-1* and significantly enhanced hypocotyl elongation in *pif457* (Fig. 5a)[63], confirming that BR biosynthesis acts downstream of PIFs[21]. The same BR treatment also rescued the hypocotyl lengths of *MIM156* and *rSPL9* to that of Col-0 (Fig. 5a), indicating that the miR156/*SPL9* module acts upstream of BR biosynthesis and is not required for BR signaling. Corroborating this conclusion, BR treatment also increased the hypocotyl length of *hyl1-2*, although, like *pif457*, the rescue was incomplete, suggesting that *hyl1-2* had minor defects in BR signaling (Fig. 5a)[21]. The discrepancies between *hyl1-2* and the *MIM156* and *rSPL9* lines were likely due to pleiotropic effects from the deficiency of other miRNAs in *hyl1-2* as, when the miRNA accumulation defects were

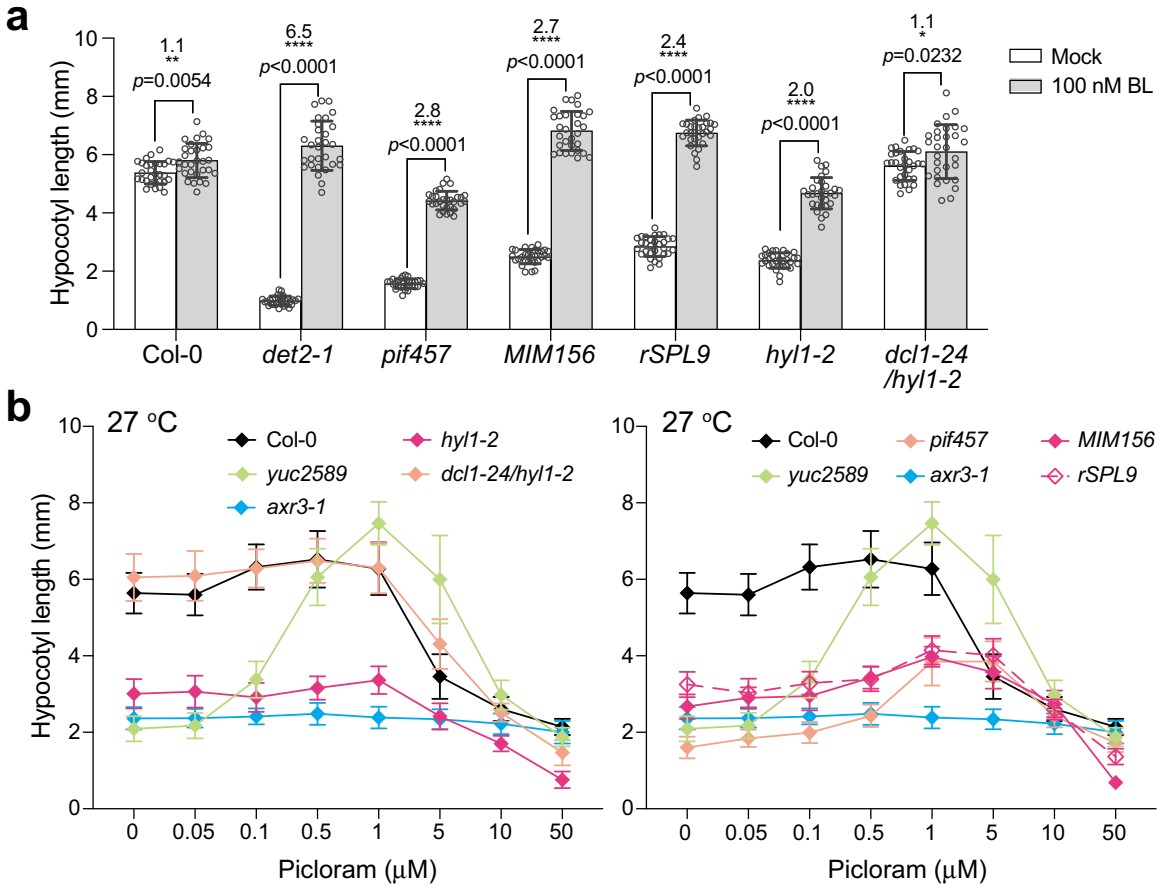

**Fig. 5 | Perturbation of miR156/SPL9 impedes auxin responsiveness at warm temperatures. a** miR156/SPL9 acts upstream of brassinosteroid biosynthesis in thermomorphogenesis. Hypocotyl length measurements of 4-d-old Col-0, *det2-1*, *pif457*, *MIM156*, *rSPL9*, *hyl1-2*, and *dcl1-24/hyl1-2* grown under 50 μmol m⁻² s⁻¹ R light at 27 °C with or without 100 nM brassinolide (BL). Error bars represent the s.d. of at least 30 seedlings. The centers of the error bars indicate the mean. The fold changes and the *p* values between the treated and mock control seedlings for each genotype are shown above the columns. Statistical significance was analyzed using two-tailed

Student's t-tests (*$p < 0.05$, **$p < 0.01$, ****$p < 0.0001$), $n$ = at least 30 seedlings. **b** Auxin dosage response curves showing that miR156 is required for the hypocotyl's responsiveness to auxin at 27 °C. Hypocotyl length measurements of 4-d-old Col-0, *yuc2589*, *axr3-1*, *pif457*, *MIM156*, *rSPL9*, *hyl1-2*, and *dcl1-24/hyl1-2* seedlings grown under 50 μmol m⁻² s⁻¹ R light at 27 °C and treated with a concentration series of picloram from 0 to 50 μM. Error bars represent the s.d., $n$ = at least 30 seedlings. The centers of the error bars indicate the mean. The source data underlying the hypocotyl measurements in (**a**) and (**b**) are provided in the Source Data file.

rescued, the suppressor *dcl1-24/hyl1-2* mutant became as tall as Col-0, which implicated full BR responsiveness (Fig. 5a).

We next tested the function of miR156 in auxin responsiveness. Auxin's effect on hypocotyl elongation is concentration-dependent: auxin promotes hypocotyl elongation at low concentrations and inhibits it at high concentrations[64]. Therefore, we treated Col-0, *pif457*, *hyl1-2*, *dcl1-24/hyl1-2*, *MIM156*, and *rSPL9* seedlings grown at 27 °C with a series of concentrations of the synthetic auxin picloram. The auxin biosynthesis mutant *yuc2589* and the auxin signaling mutant *axr3-1* were used as controls. Col-0 seedlings did not respond to low picloram concentrations ranging from 0.05 to 1 μM, consistent with an expected high endogenous auxin level at warm temperatures. As expected, *yuc2589* responded well to the low range of picloram concentrations (Fig. 5b). Both Col-0 and *yuc2589* reacted to picloram concentrations above 1 μM with reduced hypocotyl growth (Fig. 5b). In contrast, *axr3-1* was insensitive to both the low and high picloram concentrations (Fig. 5b). The *pif457* mutant also responded to both the low and high concentrations of picloram; however, its hypocotyl was far less responsive to picloram than *yuc2589*, supporting the idea that PIFs regulate both auxin biosynthesis and signaling. Interestingly, *MIM156*, *rSPL9*, and *hyl1-2* were insensitive to the low range of picloram concentrations between 0.05 and 1 μM but remained sensitive to high picloram concentrations above 1 μM (Fig. 5b). The auxin

responsiveness of *dcl1-24/hyl1-2* was normal. Together, these results strongly support the conclusion that miR156 is required for auxin responsiveness, particularly at the low concentration range of auxin.

## miR156 is also required for auxin sensitivity at lower temperatures

We then asked whether auxin responsiveness requires miR156 only at warm temperatures. To that end, we examined the auxin responsiveness of Col-0, *pif457*, *MIM156*, *rSPL9*, *hyl1-2*, and *dcl1-24/hyl1-2* to exogenous picloram at 21 °C. At the lower temperature, similar to what has been described previously[64], Col-0 showed a bell-shaped response to the low and high concentrations of picloram: auxin at low picloram concentrations between 0.05 and 5 μM stimulated hypocotyl elongation, whereas auxin at concentrations above 5 μM inhibited hypocotyl elongation (Fig. 6a). The *yuc2589* mutant only responded to 1 μM picloram and greater (Fig. 6a). *pif457* also responded to 1 μM picloram and greater but was not as responsive as Col-0 and *yuc2589* (Fig. 6a). In comparison, *MIM156*, *rSPL9*, and *hyl1-2* appeared to be the least responsive to picloram, except for *axr3-1* (Fig. 6a). Again, the auxin responsiveness defect of *hyl1-2* was completely rescued in *dcl1-24/hyl1-2* (Fig. 6a). These results indicate that miR156 is also required for auxin sensitivity at lower temperatures.

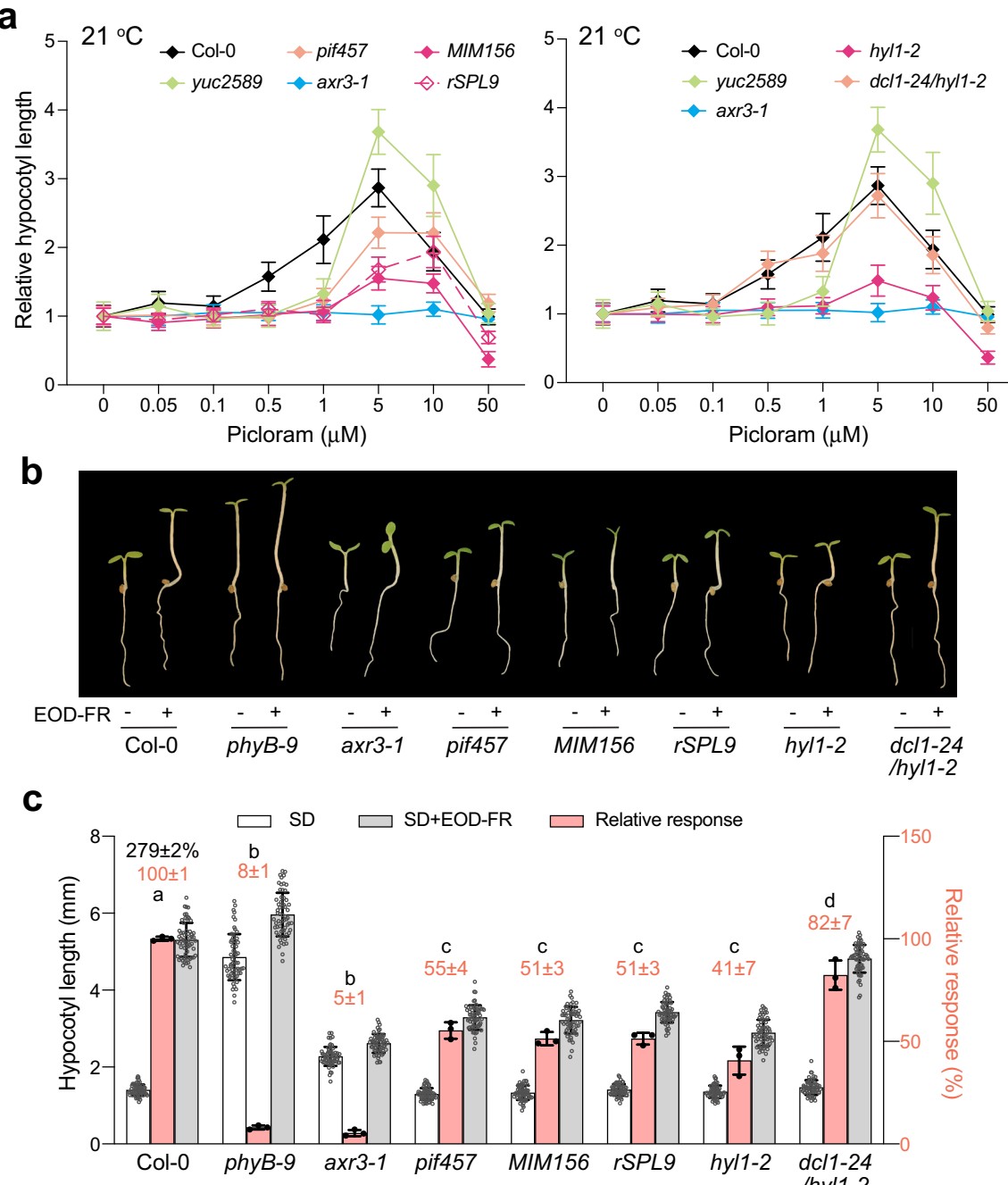

**Fig. 6 | MiR156-dependent auxin sensitivity is required for the shade avoidance response. a** Auxin dosage response curves showing that miR156 is required for the hypocotyl's responsiveness to auxin at 21 °C. Hypocotyl length measurements of 4-d-old Col-0, *yuc2589*, *axr3-1*, *pif457*, *MIM156*, *rSPL9*, *hyl1-2*, and *dcl1-24/hyl1-2* seedlings grown under 50 μmol m$^{-2}$ s$^{-1}$ R light at 21 °C and treated with a concentration series of picloram from 0 to 50 μM. Hypocotyl length was calculated relative to that without picloram treatment for each genotype. Error bars represent the s.d., *n* = at least 30 seedlings. The centers of the error bars indicate the mean. **b** Images of 4-d-old Col-0, *phyB-9*, *axr3-1*, *pif457*, *MIM156*, *rSPL9*, *hyl1-2*, and *dcl1-24/hyl1-2* seedlings grown at 21 °C under the short-day condition of 8 h of 100 μmol m$^{-2}$ s$^{-1}$ white light and 16 h of dark with or without a 15 min end-of-day FR (EOD-FR) light treatment. **c** Hypocotyl length measurements of seedlings in (**a**) and their relative response to the EOD-FR treatment. The black number above the Col-0 columns represents the percent increase in hypocotyl length (mean ± s.d., *n* = 3 biological replicates) by the EOD-FR treatment. The pink bars show the EOD-FR response of a mutant relative to that of Col-0 (set at 100%). Pink numbers show the mean ± s.d. of the relative responses. Different lowercase letters above the bars denote statistically significant differences in the relative responses (ANOVA, Tukey's HSD, *p* < 0.01, *n* = 3 biological replicates). Error bars for the relative responses represent the s.d. of three biological replicates. The centers of all error bars indicate the mean. The source data underlying the hypocotyl measurements in (**a**) and (**c**) are provided in the Source Data file.

## miR156-dependent auxin sensitivity operates in the shade avoidance response

At both high and low temperatures, hypocotyl elongation due to the shade avoidance response can be effectively stimulated by the FR light reflected from neighboring plants[16,20]. An assay simulating the

shade avoidance response is end-of-day FR (EOD-FR) treatment, in which a 15 min FR light treatment at the end of the daytime inactivates PHYB to promote hypocotyl growth during the nighttime. Like warm-temperature-dependent hypocotyl elongation, shade-induced hypocotyl elongation also relies on PIF-dependent auxin

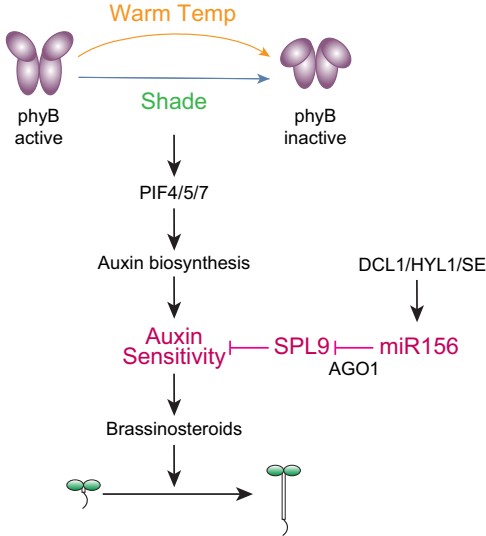

**Fig. 7 | Model for miR156-enabled phenotypic plasticity to temperature and light changes.** Environmental light and temperature changes perceived by the photoreceptor and thermosensor PHYB trigger profound modulations in plant architecture via the master growth regulators PIFs and PIF-induced auxin synthesis and signaling. This study unveils a previously unknown control of auxin sensitivity licensed by miR156. Auxin sensitivity is antagonized primarily by SPL9. In light- and temperature-elicited hypocotyl elongation responses during Arabidopsis seedling establishment, miR156 enables auxin sensitivity by repressing *SPL9*. We propose that miR156-dependent auxin sensitivity constitutes a genetic circuit gating light and temperature responses by the endogenous developmental program. This critical role of miR156 makes the basic components involved in miRNA biogenesis and function, such as DCL1, HYL1, SE, and AGO1, necessary elements for the plant's phenotypic plasticity in response to environmental temperature and light changes.

biosynthesis and signaling[16,20]. We thus asked whether miR156 is required for the auxin-dependent hypocotyl response by an EOD-FR treatment. Col-0 responded to the EOD-FR treatment with a 279% increase in hypocotyl length, and this hypocotyl response was diminished in *phyB-9* and *axr3-1* (Fig. 6b, c). The *pif457* mutant showed a 50% reduction in response relative to Col-0, as did *MIM156*, *rSPL9*, and *hyl1-2* (Fig. 6b, c). The EOD-FR response was almost completely restored in *dcl1-24/hyl1-2*. These results thus indicate that miR156 is also required for hypocotyl growth plasticity in response to light changes.

## Discussion

Plants display dramatic phenotypic plasticity in response to environmental temperature and light changes. Although it is well understood that light and temperature modulate plant development and growth prominently through the photoreceptor and thermosensor PHYB and the PHYB-dependent regulation of phytohormones, including auxin, whether and how miRNAs participate in thermomorphogenesis remained ambiguous. This study reveals miR156 and its target *SPL9* as critical regulators of thermomorphogenesis via the control of auxin sensitivity (Fig. 7). Although it has been postulated that elevated temperature downregulates the level of miR156 to accelerate floral induction[35,36], the link between the miR156/*SPL9* module, auxin sensitivity, and temperature-dependent growth plasticity has not been previously reported. Also, unlike the proposed mechanism of miR156-mediated floral reduction[35,36], miR156-dependent auxin sensitivity is not mediated by temperature-responsive miR156 accumulation, and it operates at both low and high temperatures. Thus, the miR156/*SPL9* module defines a previously uncharacterized genetic circuit that gates the integration of temperature and light cues into growth plasticity via the control of tissue/organ auxin sensitivity or the competency to grow (Fig. 7).

Our results demonstrate that the miR156/*SPL9* module is required for the hypocotyl elongation response elicited by elevated temperature and shade. This conclusion is strongly supported by the genetic evidence that *MIM156*, similar to *hyl1-2*, had a dramatically reduced response to warm temperature or shade treatments (Figs. 3d, e and 6b, c). The transcript levels of four miR156-targeted *SPLs*−*SPL3*, *SPL5*, *SPL6*, and *SPL9*−increased in *hyl1-2* and returned to wild-type levels in *dcl1-24/hyl1-2* (Fig. 3c). Among the ten miR156-regulated *SPLs*, *SPL9* appears to be the most prominent one to hinder thermoresponsive hypocotyl growth (Supplementary Fig. 1). Stabilizing *SPL9* RNA alone in the *rSPL9* line blocked hypocotyl elongation by elevated temperature and shade, providing evidence that the enhanced expression of *SPL9* caused the thermomorphogenesis defects in *hyl1-2* and *MIM156* (Figs. 3d, e and 6b, c). Therefore, miR156 enables the growth plasticity of *Arabidopsis* seedlings by repressing mainly *SPL9*. The current data do not exclude the possibility that other *SPLs* may participate in miR156-mediated temperature responses in other developmental stages or growth conditions. For instance, miR156 mediates tolerance to sustained heat stress by regulating *SPL2* and *SPL11*[65].

The level of miR156 was not altered by temperature in our experiments (Fig. 3a and Supplementary Data 1), suggesting that temperature-responsive miR156 accumulation may not be a necessary mechanism for hypocotyl growth. This conclusion is further bolstered by the essential role of miR156 in the shade avoidance response at the lower temperature (Fig. 6b, c). These findings are at odds with previous studies suggesting that warm-temperature-dependent floral induction is mediated by a significant reduction in miR156 abundance at warmer temperature[35,36]. This discrepancy may be attributable to the different developmental stages used in these studies. Our experiments were conducted using 4-d-old seedlings, where miR156 was present at high levels, while the study by Lee et al. used 10-d-old seedlings[35], where miR156 was expected to accumulate at a lower level, and therefore the effect of a reduction in pri-miR156 processing may become detectable[32,37]. It has also been shown that PIFs directly suppress the expression of *MIR156* to inhibit leaf initiation under shade conditions[39], implying that warm temperatures, through PIF4 accumulation, could also repress miR156 production at the transcriptional level. Similarly, in this latter study, miR156 was quantified at the adult stage in 4-week-old plants[39]. Therefore, it is possible that we did not observe the temperature-responsive reduction of miR156 accumulation due to the young seedling stage used in our experiments. Nonetheless, because miR156 plays a positive role in hypocotyl growth, a temperature- or shade-dependent reduction in miR156 abundance would unlikely be a major contributor to miR156-mediated hypocotyl elongation. Also, it is important to note that, despite an antagonistic relationship between PIF4/5 and miR156 in shade-induced responses in leaf number, leaf blade size and flowering time, miR156 appeared to be required for shade- or PIF4/5-induced petiole elongation at the adult stage, as *MIM156* had shorter petioles compared with the wild-type under normal light conditions and lacked the petiole elongation response in simulated shade conditions[39].

Our results unveiled an essential role for the miR156/*SPL9* module in enabling auxin sensitivity for environmentally controlled plant growth (Fig. 7). Hypocotyl elongation elicited by either warm temperatures or shade is mediated by PHYB signaling that promotes the accumulation of PIF4 and PIF7 and PIF-dependent actions of auxin and brassinosteroids[13-18]. Blocking global miRNA biogenesis in *hyl1-2* does not affect early temperature signaling leading to PIF4 accumulation (Fig. 1d), suggesting miRNAs are dispensable for early PHYB signaling events. Similar to *pif457*, the growth retardation in *MIM156*, *rSPL9*, and *hyl1-2* was rescued by exogenous brassinolide (Fig. 5a), indicating the miR156/*SPL9* module intersects with PHYB signaling at a signaling step between PIF4 accumulation and brassinosteroid biosynthesis. More importantly, the *MIM156* and *rSPL9* lines were almost insensitive to low concentrations of exogenous picloram at both low and high

temperatures (Figs. 5b and 6a), demonstrating that miR156 plays an essential role in enabling auxin responsiveness. Genome-wide transcriptome analysis suggests miR156 is required for the warm-temperature-dependent induction of auxin-responsive genes, including genes encoding auxin signaling components, such as *IAA19* and *IAA29*, as well as critical proteins for cell elongation, such as the *SAURs*, particularly the light- and temperature-inducible *SAUR19* and *SAUR26* subfamilies[59–61]. In *MIM156*, the two temperature-responsive *YUCs*, *YUC8*, and *YUC9*, were either still induced by warm temperature or expressed at high levels (Fig. 4d), suggesting that miR156 may play less of a role in regulating auxin biosynthesis, but careful assessments of auxin levels in the mutants are needed to confirm this conclusion. Intriguingly, none of the miRNAs previously shown to be relevant to the regulation of auxin signaling were identified in our study. The levels of miR393, miR160, miR167, and miR390 were all reduced in *hyl1-2* and rescued in *dcl1-24/hyl1-2* (Fig. 2e), but the expression of their target genes was not changed in these mutants, suggesting these miRNAs likely play limited roles in hypocotyl elongation at the seedling stage.

In summary, our results unveil a previously uncharacterized role of the miR156/*SPL9* module as a critical regulator that enables growth plasticity in response to changes in temperature and light (Fig. 7). Because miR156 is a well-established developmental regulator of shoot development[32], we propose that the miR156/*SPL9* module defines a developmental pathway that gates plant growth plasticity in response to environmental cues. How the miR156/*SPL9* module regulates auxin responsiveness remains unknown. SPLs could either directly regulate the expression of auxin-responsive genes or indirectly regulate auxin sensitivity through SPL-regulated genes. The current study paves the way for future investigations of the mechanism of miR156-enabled auxin sensitivity in the environmental control of plant growth.

## Methods

### Plant materials and growth conditions

The Columbia (Col-0) ecotype of *Arabidopsis thaliana* was used throughout this study. The *Arabidopsis* mutants *phyB-9*[66], *pif457*[67], *ago1-3*[68], *dcl1-20*[48], *se-1*[49], *hyl1-2*[26], and *det2-1*[69] were previously described. *pif4-2* (SAIL_1288_E07), *yuc2589* (CS69869), *axr3-1* (CS57504), *MIM156* (CS9953), *sSPL2* (CS69800), *rSPL2* (CS69801), *sSPL3* (CS69802), *rSPL3* (CS69803), *sSPL4* (CS69804), *rSPL4* (CS69805), *sSPL5* (CS69806), *rSPL5* (CS69807), *sSPL6* (CS69808), *rSPL6* (CS69809), *sSPL9* (CS698010), *rSPL9* (CS69811), *sSPL10* (CS69812), *rSPL10* (CS69813), *sSPL11* (CS69814), *rSPL11* (CS69815), *sSPL13* (CS69816), *rSPL13* (CS69817), *sSPL15* (CS69818), *rSPL15* (CS69819), *spl9-4* (CS67866), *spl2/9/10/11/13/15* (CS69799), and *spl3/4/5* (CS69790) were obtained from the Arabidopsis Biological Resource Center. The *pif4-2/hyl1-2* and *dcl1-24/hyl1-2* mutants were generated in this study.

*Arabidopsis* seeds were surface-sterilized and plated on half-strength Murashige and Skoog (½ MS) medium containing Gamborg's vitamins (Caisson Laboratories), 0.5 mM MES pH 5.7, and 0.8% agar (w/v). Seeds were stratified in the dark at 4 °C for 5 days before treatment under specific light and temperature conditions in LED chambers (Percival Scientific, Perry, IA). Fluence rates of light were measured with an Apogee PS200 spectroradiometer (Apogee Instruments, Logan, UT) and SpectraWiz spectroscopy software (StellarNet, Tampa, FL). For the warm temperature treatment, seedlings were grown under 50 µmol m$^{-2}$ s$^{-1}$ R light at 21 °C for 2 days (48 h) and then either kept in the same conditions or transferred to 27 °C under the same light conditions for two additional days (48 h). To minimize the influence of the circadian clock, all phenotypic characterization, such as hypocotyl measurements, and all molecular characterization, including global miRNA and transcriptome analyses, were performed 96 h after stratification. For the 21-to-27 °C transition experiments, 4-d-old seedlings grown under 50 µmol m$^{-2}$ s$^{-1}$ R light at 21 °C were transferred to 27 °C

under the same light conditions for the indicated time period. For end-of-day far-red (EOD-FR) light treatments, seedlings were grown in 100 µmol m$^{-2}$ s$^{-1}$ white light under short-day conditions (8 h light/16 h dark) and treated with 10 µmol m$^{-2}$ s$^{-1}$ FR light for 15 min at the end of the day. Detailed growth conditions are provided in the respective figure legends.

### Hypocotyl measurements

For hypocotyl measurements, at least 30 seedlings from each genotype were scanned using an Epson Perfection V700 photo scanner, and hypocotyl length was measured using NIH ImageJ software (http://rsb. info.nih.gov/nih-image/). The warm-temperature response was assessed as the percent increase in hypocotyl length of each genotype at 27 °C relative to 21 °C. The relative response was calculated using the percent increase in hypocotyl length of a mutant divided by that of Col-0. At least three replicates were used to calculate the mean and standard deviation of the relative response. Bar charts were generated using Prism 8 (GraphPad Software, San Diego, CA). Images of representative seedlings were taken using a Leica MZ FLIII stereo microscope (Leica Microsystems Inc., Buffalo Grove, IL) and processed using Adobe Photoshop v22.3.0 (Adobe Inc., Mountain View, CA).

### EMS mutagenesis and *hyl1-2* suppressor screen

*hyl1-2* seeds were mutagenized with EMS[17]. First, 0.2 g of *hyl1-2* seeds were soaked in 45 ml of ddH$_2$O with 0.005% Tween-20 for 4 h, washed twice with ddH$_2$O, and then soaked in 0.2% EMS (MilliporeSigma, St. Louis, MO) for 15 h with rotation. Subsequently, the seeds were washed with ddH$_2$O eight times, stratified in the dark at 4 °C for 4 days, and sown onto ½ MS plates. A total of 1000 M1 seedlings were randomly selected and grown to flowering. M2 seeds were collected from each M1 plant individually. We then performed family screening for *hyl1-2* suppressors using the M2 seeds from the 1000 families. For the primary screen, at least eighty M2 seeds from each M1 family were grown under 50 µmol m$^{-2}$ s$^{-1}$ R light at 21 °C for 2 days and at 27 °C for two additional days, and putative suppressors with a longer hypocotyl than *hyl1-2* were selected and subjected to a secondary screen for mutants with a hypocotyl length similar to *hyl1-2* at 21 °C−i.e., we screened for mutants with a longer hypocotyl than *hyl1-2* at only 27 °C, not at 21 °C.

### Mapping point mutations

The *hs400/hyl1-2* suppressor mutant was backcrossed to *hyl1-2*. Genomic DNA was isolated from either 96 B2 seedlings with a *hyl1-2*-like short-hypocotyl phenotype or homozygous *hs400/hyl1-2* seedlings in the M3 generation as described previously[70]. A total of 0.25 g seedlings were ground in liquid nitrogen, and the powder was resuspended in 30 ml of ice-cold extraction buffer 1 (0.4 M sucrose, 10 mM Tris, pH 8, 10 mM MgCl$_2$, 5 mM β-mercaptoethanol) and filtered through Miracloth. The sample was centrifuged at 1940 × *g* for 20 min at 4 °C. The pellet was resuspended in 1 ml of chilled extraction buffer 2 (0.25 M sucrose, 10 mM Tris, pH 8, 10 mM MgCl$_2$, 1% Triton X-100, 5 mM β-mercaptoethanol) and spun at 12,000 × *g* for 10 min at 4 °C. The pellet was resuspended in 300 µl of chilled extraction buffer 3 (1.7 M sucrose, 10 mM Tris, pH 8, 2 mM MgCl$_2$, 0.15% Triton X-100, 5 mM β-mercaptoethanol) and overlaid on top of 300 µl of chilled extraction buffer and spun at 14,000 × *g* for 1 h at 4 °C. The pellet, which contained the nuclei, was resuspended in 1.5 ml of CTAB extraction buffer (100 mM Tris-HCl pH 8.0, 1.4 M NaCl, 20 mM EDTA pH 8.0, 2% CTAB) and incubated for 30 min at 60 °C. Then, 1 µl RNase A (20 mg/ml stock solution) was added to the supernatant to remove RNA. Genomic DNA was extracted with chloroform/isopentanol (24:1) and precipitated with isopropanol with centrifugation at 20,000 × *g* for 10 min at 4 °C. The DNA pellet was washed with 75% ethanol and resuspended in water.

Whole-genome sequencing was performed by Novogene Corporation Inc. (Sacramento, CA). The genomic DNA was randomly

sheared into short fragments, and the obtained fragments were end-repaired, A-tailed, and further ligated to Illumina adapters. The fragments with adapters were PCR amplified, size selected, and purified. The library was quantified with Qubit and real-time PCR, examined for size distribution with a bioanalyzer, and then subjected to paired-end 150 bp sequencing using NovaSeq 6000 (Ilumina, Inc., San Diego, CA) to generate 5 Gb of data. The *hs400* mutation was identified using the SIMPLE[53] pipeline by treating the M3 *hs400/hyl1-2* sample as the mutant and the pooled *hyl1-2*-like B2 sample as the wild type. The *dcl1-24* allele was confirmed using the following dCAPS (derived cleaved amplified polymorphic sequences) marker with the PCR primers: dcl1-F (CCTCAAAAGCACGAGGGTCA) and dcl1-R (TCCATCTTTTGTGTCCTCGTCGAAAATCG); digesting the PCR product with the restriction enzyme Hpy188I (New England BioLabs, Ipswich, MA) yields one 180-bp fragment for *dcl1-24* and two fragments of 152-bp and 28-bp for Col-0.

### RNA extraction and quantitative real-time PCR
Seedlings (50–100 mg) were collected by flash freezing in liquid nitrogen and stored at −80 °C until processing. Samples were ground to a fine powder in liquid nitrogen, and RNA was extracted using a Quick-RNA MiniPrep kit with on-column DNase I digestion (Zymo Research, Irvine, CA). cDNA synthesis was performed with 1 μg total RNA using a Superscript II First Strand cDNA Synthesis Kit (Thermo Fisher Scientific, Waltham, MA). For qRT-PCR, cDNA diluted in nuclease-free water was mixed with iQ SYBR Green Supermix (Bio-Rad Laboratories, Hercules, CA) and gene-specific primers (Supplementary Table 1). qRT-PCR reactions were performed using a Roche LightCycler 96 system. Transcript levels were calculated relative to the level of *PP2A*. Bar charts were generated using Prism 8 (GraphPad Software, San Diego, CA).

### Small RNA-seq and RNA-seq analysis
For small RNA-seq, 10 μg of total RNA from 4-d-old seedlings grown at either 21 °C or 27 °C was resolved on a 15% urea-PAGE gel, the 15–40-nt region was excised, and small RNAs were purified from the gel. Small RNA libraries were constructed using a NEBNext Multiplex Small-RNA Library Prep Set for Illumina (E7300). The libraries were sequenced by Novogene Corporation Inc. (Sacramento, CA) using HiSeq X (Ilumina, Inc., San Diego, CA). The sequencing data were analyzed using the homemade pipeline pRNASeqtools v0.8 (https://github.com/grubbybio/pRNASeqTools). The raw reads were processed to remove the 3′ adapter sequence using cutadapt v3.0[71]. Trimmed reads of 18–42-nt were aligned to the *Arabidopsis* genome (Araport11)[72] using ShortStack v3.8[73] with parameters (--mismatches 0 --mmap u --bowtie_m 1000 --ranmax 50). Reads were normalized to 45S rRNA reads and RPMR (reads per million of 45S rRNA reads) values were calculated. Differential miRNAs were identified using DESeq2 v1.35.0[74]. Z-scores were computed on a gene-by-gene (row-by-row) basis by subtracting the mean and then dividing by the standard deviation. The computed Z score is then used to plot the heatmap.

RNA-seq sequencing was performed by Novogene Corporation Inc. (Sacramento, CA). Messenger RNA was isolated from total RNA using poly-T oligo-attached magnetic beads. After fragmentation, first-strand cDNA was synthesized using random hexamer primers, followed by second-strand cDNA synthesis. The libraries were subjected to paired-end 150 bp sequencing using NovaSeq 6000 (Ilumina, Inc., San Diego, CA). RNA-seq data analysis was performed using the homemade pRNASeqTools pipeline (https://github.com/grubbybio/pRNASeqTools). Briefly, the raw reads were mapped to the Araport11[72] genome using STAR version 2.7.7a[75], and the number of reads mapped uniquely to each annotated gene was counted using featureCounts version 2.0.1[76]. Transcript levels were measured in fragments per kilobase per million total mapped fragments (FPKM). Differentially expressed genes were identified using

DEseq2 v1.35.0[74] with a fold change of two and $p < 0.01$ as the parameters.

Violin plots were generated using ggpubr v0.4.0[77]. Heatmaps were generated using pheatmap v1.0.12[78]. Volcano plots were generated using EnhancedVolcano v1.17.0[79]. Venn diagrams were generated using ggvenn v0.1.9[80]. GO cluster analysis was performed using clusterProfiler 4.0[58].

### Protein extraction and immunoblots
Total protein from 100 mg of 4-day-old seedlings was extracted in 300 μl of extraction buffer containing 100 mM Tris-Cl pH 7.5, 100 mM NaCl, 5 mM EDTA, 5% SDS, 20% glycerol, 20 mM DTT, 40 mM β-mercaptoethanol, 2 mM phenylmethylsulfonyl fluoride, 40 μM MG115, 40 μM MG132, 10 mM N-ethylmaleimide, 1× phosphatase inhibitor cocktail 3 (MilliporeSigma, Burlington, MA), 1× EDTA-free protease inhibitor cocktail (MilliporeSigma, Burlington, MA), and 0.01% bromophenol blue. Samples were immediately boiled for 10 min and centrifuged at $16,000 \times g$ for 10 min. Cleared protein samples were separated via SDS-PAGE, transferred to nitrocellulose membranes, probed with the indicated primary antibodies, and then incubated with a 1:5000 dilution of horseradish peroxidase-conjugated goat anti-rabbit or anti-mouse secondary antibodies (Bio-Rad Laboratories, 1706515 for anti-rabbit and 1706516 for anti-mouse). Primary antibodies, including polyclonal rabbit anti-PIF4 antibodies (Agrisera, AS12 1860) and monoclonal mouse anti-actin antibodies (Sigma-Aldrich, A0480), were used at a 1:2000 dilution. The signals were detected via a chemiluminescence reaction using SuperSignal West Dura Extended Duration Substrate (ThermoFisher Scientific, Waltham, MA).

### Reporting summary
Further information on research design is available in the Nature Portfolio Reporting Summary linked to this article.

## Data availability
The *Arabidopsis* mutants used in the current study are available from the corresponding authors upon request. The small RNA and RNA sequencing data associated with this study have been deposited to the GEO database with the accession code GSE216362. The source data underlying Figs. 1b–d, 2b, 3e, 5a, b, 6a, 6c, and Supplementary Fig. 1 are provided as a Source data file. Source data are provided with this paper.

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

## Acknowledgements

We thank Christian Fankhauser (University of Lausanne) for sharing the *pif457* mutant and Joanne Chory (Salk Institute) for providing the *det2-1* mutant. We thank Guy Wachsman from Philip Benfey's lab at Duke University for his kind assistance with the SIMPLE point mutation mapping pipeline. We are grateful to Elise Pasoreck for valuable comments on the manuscript. This work was supported by National Institute of General Medical Sciences grants R01GM129373 to X.C. and R01GM087388 to M.C. Q.S. was supported partially by a postdoc fellowship from Shenzhen University, China.

## Author contributions

Q.S., L.F., Y.Q., B.M., M.C., and X.C. conceived of the original research plan; M.C. and X.C. supervised the experiments; Q.S., L.F., T.L., Y.Q., J.D., and M.C. performed the experiments; Q.S., L.F., T.L., Y.Q., J.D., B.M., M.C., and X.C. analyzed the data; Q.S., L.F., T.L., Y.Q., J.D. B.M., M.C., and X.C. wrote the article.

## Competing interests

The authors declare no competing interests.
