## [Peer Review File · Nature Communications]

MicroRNA156 conditions auxin sensitivity to enable growth plasticity in response to environmental changes in ArabidopsisReviewer #1 (Remarks to the Author):

In this manuscript, authors explored the function of miRNAs in thermomorphogenesis. In particular, they identified miR156 and its targets SPL transcription factors as critical regulators of thermomorphogenesis. These results are novel and interesting. The paper is well written and easy to follow. Therefore, I am general positive on this manuscript.

One major concern is the conclusion "SPL9 is the major member of the SPL gene family involved in thermomorphogenesis" in the Abstract is not well justified. Authors used MIM156 and rSPL9 throughout the manuscript. I could not find any data related to the spl9 mutant. It is very likely that other miR156-targeted SPLs also contribute to the high temperature-induced cell elongation. This point has to be clarified in the revision.

Reviewer #2 (Remarks to the Author):

Plants constantly adjust their growth in response to variations of environmental parameters. In many species like *Arabidopsis thaliana*, mild elevated temperatures trigger a developmental response known as thermomorphogenesis. It is notably characterized by increased elongation of vegetative organs (hypocotyl, petioles, roots, etc.), hyponastic leaves as well as early flowering at later stages of development. Elevation of temperature is sensed by the red/far red-sensing phytochrome B photoreceptor. The signaling cascade goes through PIF transcription factors and then leads to the production of auxin, which allows hypocotyl elongation.

In the present study, the authors aim at studying the potential role of miRNAs in thermomorphogenesis at young seedling stage and identify key actors. They first observed that some mutants affected in miRNA biogenesis, in particular hyl1-2 and ago1-3, show defects in hypocotyl elongation in response to elevated temperatures. Thanks to a forward genetic screen, they found suppressor (dominant) mutations that restored the thermomorphogenic response of the hyl1-2 mutant: all three of them in DCL1, and strikingly similar to a previously described dcl1-24 allele. When comparing miRNA levels in WT, hyl2-1 and dcl1-24hyl2-1 seedlings at 27°C, they found 56 miRNA downregulated in hyl2-1 compared to WT that were also (partially) rescued in the double mutant. This suggests that the defective response to temperature in hyl1-2 was due to a defect in miRNA biogenesis. As they could not find any temperature-regulated miRNA, they looked for differentially expressed miRNA targets (criteria: up-regulated in hyl2-1 compared to WT at 27°C, with a rescue in the double mutant) and found, among others, four SPL genes. They chose to test the role of miR156 and SPL9 in thermomorphogenesis by using both MIM156 (target mimicry, blocking miR156 activity) and rSPL9 (miR-resistant SPL9 transgene). Both lines have reduced elongation at 27, showing the role of miR156 and SPL9 in hypocotyl elongation in response to temperature. Different experiments (RNA-seq, pharmacological experiments, phenotyping), they concluded that miR156 is required for auxin sensitivity (and not brassinosteroid signaling) at both 21°C and 27°C. Finally, the miR156/SPL9 module, with mir156 biogenesis controlled partially by HYL1, appears to be a parallel pathway to the classical phyB/PIF/auxin regulon in thermomorphogenesis. It regulates auxin sensitivity apparently independently of the perception of the temperature (or light) initial signal and serves as an endogenous way to control plasticity.

This study brings exciting new insights on the control of growth responses to variations of the environment, by highlighting new actors that adjusts auxin sensitivity independently of known signaling players like PIFs. It is well conducted and the manuscript is well written.

Here are a few comments to further improve the manuscript.

1. It seems that this study brings "contradictory" results with what has been published before on the role of miR156 (differentially expressed in response to elevated temperatures at other stages of development, expression directly suppressed by PIF4/PIF5 under shade conditions). Indeed, experiments performed at different stages of development, under potentially different experimental conditions (light, temperature, etc) can lead to different results. I was wondering whether the authors have considered testing hyl1-2, MIM156 and rSPL9 phenotypes at later stages of development, for example petiole elongation on young rosettes. Whatever the results it might

bring good arguments to reinforce the role of miR156 highlighted in this study.

2. Lines 165-166: "pif4-2 could be considered as a weak pif457 allele, with hyl1-2 enhancing its phenotype to that of the strong pif457 mutant." I do not fully agree with this assertion, because it would mean that PIF4, PIF5 and PIF7 have the exact same function, which is not the case. I am more convinced by the fact that HYL1 and PIF4 act in parallel pathways.

3. When testing the phenotype of ago1-3 mutant, the authors highlight the fact that it has pleiotropic developmental defects. Could it be relevant to test another allele of ago1-3 to confirm the phenotype, maybe a less drastic one (if such allele exists)?

4. When performing the forward genetic screen, the authors found 3 suppressors carrying a mutation in DCL1 leading to a proline to serine substitution (P1448S), similar to the one found in dcl1-24. In the article from 2012 (ref. 80), they mapped a C to T mutation. Is it the same exact mutation found in hs400, hs470 and hs471?

Also, it is not clear whether the authors used the dcl1-24/hyl1-2 double mutant previously described (ref. 80) or one of the suppressor lines generated in this study. Even if the mutation in DCL1 is similar, the genetic background might be slightly different (because of independent propagation of seeds for example).

5. Lines 215-216: the sentence "Therefore, we concluded that thermomorphogenesis requires the action of one or more of the 56 rescued miRNAs in dcl1-24/hyl1-2" is not an appropriate conclusion at that stage. The comparison of miRNA levels between the different genotypes is a good evidence for the fact that the phenotypic defect in hyl1-2 at elevated temperature is due to a deficiency in miRNA biogenesis. I would suggest to rephrase this part.

6. In the final model (figure 7), the authors have put both miR156 and SPL9 as a parallel pathway to phyB/PIF/etc as described. I was wondering why not putting also HYL1 on the model, because this is how they finally found miR156 (even if I agree that miR156 function does not explain the full effect of HYL1).

Reviewer #3 (Remarks to the Author):

Sang et al explores the roles of miRNA in regulating plant thermomorphogenesis. In previous studies, mainly the light signaling pathways have been shown to be involved in regulating thermomorphogenesis. The red light photoreceptor, phyB, and the signaling factor PIF4 function as a high ambient temperature sensor and a critical regulator, respectively. In this story, the authors show that miRNA156/SPL9 module functions in regulating thermomorphogenesis in concert with PIF4. They also show that this module regulates thermomorphogenesis under both light and shade conditions by controlling auxin sensitivity, and not by auxin biosynthesis. Overall, this study identifies a new player regulating thermomorphogenesis. The experiments are designed and executed well and the results support their conclusions. A couple of minor comments to improve the manuscript:

1. Through various analysis, they identified miRNA156/SPL9 module as key regulator of thermomorphogenesis under both light and shade conditions. Although they produced MIM156 and rSPL9 genetic materials to assess the role of this module, the function of SPL9 in this pathway would be better assessed using a spl9 mutant background and perhaps with other spl higher order mutants (if available, no need to make for this story). This would highlight the roles, if any, of other miRNA156 targets in this pathway.

2. Lines 24-26, the conclusion that HYL1 acts downstream of PIF4 is not supported by the data. In fact, the fig. 1a shows that the hyl1pif4 double looks more like pif457, suggesting that PIFs might be acting downstream of HYL1.

3. Although they cited the papers showing phyB as a temperature sensor, the recent story showing phyB can sense high ambient temperature using the N-terminal extension through phase

separation can also be cited and discussed in this context (Chen, D., et al. (2022). "Integration of light and temperature sensing by liquid-liquid phase separation of phytochrome B." *Molecular Cell* 82: 3015-3029).

Response to Reviewers

We thank all reviewers for the positive comments and valuable suggestions.

Reviewer #1

Reviewer #1 (Remarks to the Author):

In this manuscript, authors explored the function of miRNAs in thermomorphogenesis. In particular, they identified miR156 and its targets SPL transcription factors as critical regulators of thermomorphogenesis. These results are novel and interesting. The paper is well written and easy to follow. Therefore, I am general positive on this manuscript.

One major concern is the conclusion “SPL9 is the major member of the SPL gene family involved in thermomorphogenesis” in the Abstract is not well justified. Authors used MIM156 and rSPL9 throughout the manuscript. I could not find any data related to the spl9 mutant. It is very likely that other miR156-targeted SPLs also contribute to the high temperature-induced cell elongation. This point has to be clarified in the revision.

Response: We thank the reviewer for the comments. We now included new data on the thermomorphogenic responses of the miR156-sensitive and miR156-resistant lines for all ten miR156-regulated SPLs in Supplementary Fig. 1a. The new data show that *SPL9* is the only gene whose miR156-resistant line (*rSPL9*) exhibited a dramatic reduction in thermoresponsive hypocotyl elongation compared with the miR156-sensitive line (*sSPL9*). Although *rSPL2* and *rSPL3* also had a slight reduction in the thermal response compared to their corresponding *sSPL* lines, their phenotypes were minor compared to *rSPL9*. Our new results demonstrate that miR156 promotes thermoresponsive hypocotyl growth mainly via *SPL9*.

We also characterized the thermal responses of the *spl9-4*, *spl2/9/10/11/13/15*, and *spl3/4/5* mutants. These mutants did not display a significant phenotype (Supplementary Fig. 1b), suggesting miR156 can effectively suppress the function *SPL9* in our assay conditions.

Reviewer #2

Reviewer #2 (Remarks to the Author):

This study brings exciting new insights on the control of growth responses to variations of the environment, by highlighting new actors that adjusts auxin sensitivity independently of known signaling players like PIFs. It is well conducted and the manuscript is well written.

Here are a few comments to further improve the manuscript.

1. It seems that this study brings “contradictory” results with what has been published before on the role of miR156 (differentially expressed in response to elevated temperatures at other stages of development, expression directly suppressed by PIF4/PIF5 under shade conditions). Indeed, experiments performed at different stages of development, under potentially different experimental conditions (light, temperature, etc) can lead to different results. I was wondering whether the authors have considered testing *hyl1-2*, *MIM156* and *rSPL9* phenotypes at later stages of development, for example petiole elongation on young rosettes. Whatever the results it might bring good arguments to reinforce the role of miR156 highlighted in this study.

Response: We thank the reviewer for the comment. A previous report by Xie et al. [Nat Commun (2017) 8:348] has thoroughly characterized the relationship between PIF4/5 and miR156 in the shade avoidance response at the adult stage (4-week-old plants). Their results showed an antagonistic relationship between PIF4/5 and miR156 in shade-regulated leaf number, leaf blade size, and flowering time. However, miR156 appears to be required for shade- or PIF4/5-induced petiole elongation at the adult stage, as *MIM156* plants had shorter petioles compared with the wild-type under normal light conditions and lacked the petiole elongation response in simulated shade conditions [Fig. 2b, Nat Commun (2017) 8:348]. These results are consistent with our conclusion on the role of miR156 in PIF4-mediated hypocotyl elongation at the seedlings stage (which was not investigated by Xie et al.).

2. Lines 165-166: “*pif4-2* could be considered as a weak *pif457* allele, with *hyl1-2* enhancing its phenotype to that of the strong *pif457* mutant.” I do not fully agree with this assertion, because it would mean that PIF4, PIF5 and PIF7 have the exact same function, which is not the case. I am more convinced by the fact that HYL1 and PIF4 act in parallel pathways.

Response: We thank the reviewer for the comment. We have revised this part to the following: “The *hyl1-2/pif4-2* mutant was slightly less responsive to the warm temperature than the single *hyl1-2* and *pif4-2* mutants (Fig. 1a,b), suggesting that HYL1 and PIF4 work in both overlapping and parallel pathways. One possibility is that the function of HYL1 is required for the action of multiple PIFs, as the phenotype of *hyl1-2/pif4-2* was similar to *pif457* (Fig. 1a,b). However, we could not exclude the possibility that HYL1 is also involved in PIF-independent pathways.”

3. When testing the phenotype of *ago1-3* mutant, the authors highlight the fact that it has pleiotropic developmental defects. Could it be relevant to test another allele of *ago1-3* to confirm the phenotype, maybe a less drastic one (if such allele exists)?

Response: We did test another fertile hypomorphic allele *ago1-27* [Morel J. (2002) *Plant Cell* 14:629-39]. As shown in the following figure, *ago1-27* showed only a weak phenotype in thermoresponsive hypocotyl elongation compared with *ago1-3*. This result is consistent with the observation that the thermomorphogenesis phenotype was obvious in the null *hyl1-2* mutant but less prominent in the hypomorphic *dcll-20* and *se-1* (Fig. 1a,b).

4. When performing the forward genetic screen, the authors found 3 suppressors carrying a mutation in *DCL1* leading to a proline to serine substitution (P1448S), similar to the one found in *dcl1-24*. In the article from 2012 (ref. 80), they mapped a C to T mutation. Is it the same exact mutation found in *hs400*, *hs470* and *hs471*?

Also, it is not clear whether the authors used the *dcl1-24/hyl1-2* double mutant previously described (ref. 80) or one of the suppressor lines generated in this study. Even if the mutation in *DCL1* is similar, the genetic background might be slightly different (because of independent propagation of seeds for example).

Response: We thank the reviewer for the comments. All three suppressor mutants from our genetic screen carried the same mutation as the previously described *dcl1-24*. We renamed *hs400/hyl1-2* to *dcl1-24/hyl1-2* and used this suppressor mutant for the analysis in our study. To clarify these points, we revised the text to the following:

“Using a point mutation mapping pipeline called SIMPLE⁵⁶, we identified a proline-to-serine substitution at amino acid 1448 (P1448S) in *DCL1* in *hs400/hyl1-2* (Fig. 2c). Interestingly, this exact mutant allele of *DCL1* was previously isolated as *dcl1-24* from a different *hyl1-2* suppressor screen based on the rescue of the leaf hyponasty phenotype in adult *hyl1-2* plants⁵⁷. We then sequenced the *DCL1* locus of *hs470/hyl1-2* and *hs471/hyl1-2* and found that these two suppressor mutants also carried the same mutation as in the previously described *dcl1-24*. The *hs400/hyl1-2* mutant, designated *dcl1-24/hyl1-2* hereafter, was used for the subsequent analysis.”

5. Lines 215-216: the sentence “Therefore, we concluded that thermomorphogenesis requires the action of one or more of the 56 rescued miRNAs in *dcl1-24/hyl1-2*” is not an appropriate conclusion at that stage. The comparison of miRNA levels between the different genotypes is a good evidence for the fact that the phenotypic defect in *hyl1-2* at elevated temperature is due to a deficiency in miRNA biogenesis. I would suggest to rephrase this part.

Response: We revised the sentence to: “Thus, the *dcl1-24/hyl1-2* suppressor mutant provides genetic evidence supporting that the thermomorphogenesis defect in *hyl1-2* is due to the deficiency in miRNA biogenesis.”

6. In the final model (figure 7), the authors have put both miR156 and SPL9 as a parallel pathway to phyB/PIF/etc as described. I was wondering why not putting also HYL1 on the model, because this is how they finally found miR156 (even if I agree that miR156 function does not explain the full effect of HYL1).

Response: This is a good suggestion. We added DCL1/HYL1/SE and AGO1 to the model.

Reviewer #3

Reviewer #3 (Remarks to the Author):

Sang et al explores the roles of miRNA in regulating plant thermomorphogenesis. In previous studies, mainly the light signaling pathways have been shown to be involved in regulating thermomorphogenesis. The red light photoreceptor, phyB, and the signaling factor PIF4 function as a high ambient temperature sensor and a critical regulator, respectively. In this story, the authors show that miRNA156/SPL9 module functions in regulating thermomorphogenesis in concert with PIF4. They also show that this module regulates thermomorphogenesis under both light and shade conditions by controlling auxin sensitivity, and not by auxin biosynthesis. Overall, this study identifies a new player regulating thermomorphogenesis. The experiments are designed and executed well and the results support their conclusions. A couple of minor comments to improve the manuscript:

1. Through various analysis, they identified miRNA156/SPL9 module as key regulator of thermomorphogenesis under both light and shade conditions. Although they produced MIM156 and rSPL9 genetic materials to assess the role of this module, the function of SPL9 in this pathway would be better assessed using a *spl9* mutant background and perhaps with other *spl* higher order mutants (if available, no need to make for this story). This would highlight the roles, if any, of other miRNA156 targets in this pathway.

Response: We thank the reviewer for the comments. We have included new results in Supplementary Fig. 1b to show that the *spl9-4*, *spl2/9/10/11/13/15*, and *spl3/4/5* mutants had thermo-responsive hypocotyl responses similar to that of Col-0. We examined these three lines because previous studies showed that the ten miR156-regulated SPLs could be divided into three groups: (1) *SPL2*, *SPL9*, *SPL10*, *SPL11*, *SPL13*, and *SPL15* control both the juvenile-to-adult vegetative transition and the vegetative-to-reproductive transition; (2) *SPL3*, *SPL4* and *SPL5* mainly promote floral meristem identity transition; (3) *SPL6* does not play a major role in shoot morphogenesis [Xu et al. (2016) *Plos Genet* 12:e1006263]. We therefore examined the *spl9-4* single mutant and the higher order mutants of the first two groups of SPLs. We also characterized the phenotypes of the miR156-sensitive and miR156-resistant lines for all ten miR156-regulated

SPLs (Supplementary Fig. 1a). The new data demonstrate that miR156 control thermoresponsive hypocotyl growth by repressing mainly *SPL9*.

2. Lines 24-26, the conclusion that *HYL1* acts downstream of *PIF4* is not supported by the data. In fact, the fig. 1a shows that the *hyl1pif4* double looks more like *pif457*, suggesting that *PIFs* might be acting downstream of *HYL1*.

Response: The conclusion that *HYL1* acts downstream of *PIF4* are supported by the following lines of evidence: (1) the phenotype of the *hyl1-2/pif4-2* double mutant suggests that *HYL1* and *PIF4* act in overlapping and parallel pathways; (2) the *hyl1-2* mutant had a normal temperature-induced accumulation of *PIF4* mRNA and protein, suggesting that *HYL1* regulates a temperature signaling step downstream of *PIF4* accumulation; (3) the hypocotyl growth defect of *hyl1-2* and *MIM156* could be rescued by exogenous brassinolide (BR), therefore, *HYL1* and miR156 function upstream of BR in the *PIF4* pathway; (4) *HYL1* and miR156 are required for auxin responsiveness that has been demonstrated to be downstream of *PIF4* and *PIF4*-mediated auxin biosynthesis.

3. Although they cited the papers showing *phyB* as a temperature sensor, the recent story showing *phyB* can sense high ambient temperature using the N-terminal extension through phase separation can also be cited and discussed in this context (Chen, D., et al. (2022). "Integration of light and temperature sensing by liquid-liquid phase separation of phytochrome B." *Molecular Cell* 82: 3015-3029).

Response: We cited the two references demonstrating *phyB* as thermosensors to support our investigations of the function of miRNAs in the context of *phyB*-mediated thermomorphogenesis. The paper by Chen et al. suggested a new temperature-sensing mechanism by *phyB*. We did not cite this reference because it is not pertinent to the focus of this study, which reveals a new role of miR156 in downstream temperature signaling by auxin. We had to remove more than 20 references in the revised manuscript (many are closely relevant to this study) because we exceeded the maximum number of references allowed by the journal – 80 references.

Reviewer #1 (Remarks to the Author):

My concerns have been successfully addressed. I am happy to recommend it for publication in Nat Commun.

Reviewer #2 (Remarks to the Author):

In this revised manuscript and the accompanying response to reviewers, the authors have appropriately addressed and discussed all the comments I had raised. Besides, by responding to comments from the other reviewers, they have further improved their study.

Reviewer #3 (Remarks to the Author):

My concerns have been adequately addressed. It is a much better story now.